# Plasticity in visual cortex is disrupted in a mouse model of tauopathy

Amalia Papanikolaou [1,3 ✉], Fabio R. Rodrigues[1,3], Joanna Holeniewska[1], Keith G. Phillips[2], Aman B. Saleem[1,4] & Samuel G. Solomon [1,4]

Alzheimer's disease and other dementias are thought to underlie a progressive impairment of neural plasticity. Previous work in mouse models of Alzheimer's disease shows pronounced changes in artificially-induced plasticity in hippocampus, perirhinal and prefrontal cortex. However, it is not known how degeneration disrupts intrinsic forms of brain plasticity. Here we characterised the impact of tauopathy on a simple form of intrinsic plasticity in the visual system, which allowed us to track plasticity at both long (days) and short (minutes) time-scales. We studied rTg4510 transgenic mice at early stages of tauopathy (5 months) and a more advanced stage (8 months). We recorded local field potentials in the primary visual cortex while animals were repeatedly exposed to a stimulus over 9 days. We found that both short- and long-term visual plasticity were already disrupted at early stages of tauopathy, and further reduced in older animals, such that it was abolished in mice expressing mutant tau. Additionally, visually evoked behaviours were disrupted in both younger and older mice expressing mutant tau. Our results show that visual cortical plasticity and visually evoked behaviours are disrupted in the rTg4510 model of tauopathy. This simple measure of plasticity may help understand how tauopathy disrupts neural circuits, and offers a translatable platform for detection and tracking of the disease.

---

[1] UCL Institute of Behavioural Neuroscience, Department of Experimental Psychology, University College London, London WC1H 0AP, UK. [2] Eli Lilly, Research and Development, Erl Wood, Surrey GU20 6PH, UK. [3] These authors contributed equally: Amalia Papanikolaou, Fabio R. Rodrigues. [4] These authors jointly supervised this work: Aman B. Saleem, Samuel G. Solomon. ✉email: amalia.papanikolaou@ucl.ac.uk

Neurodegenerative diseases are known to affect neural plasticity and memory. Previous research has identified cellular and circuit deficits in higher-order brain areas such as the entorhinal and hippocampal cortices, which may underlie the deficits in memory function. For example, in vitro measurements show artificially-induced (usually electrical) long term potentiation (LTP) is reduced in the hippocampus and frontal cortex in mouse models of Alzheimer's Disease[1–4]. In vivo measurements in these mouse models also show disruption of artificially-induced LTP[5,6]. The complex connections and functions of the hippocampus and frontal cortex, however, make it difficult to design experiments that measure plasticity in these areas without using artificial stimulation. How degeneration disrupts intrinsic forms of brain plasticity is, therefore, largely unknown.

Plasticity and neurodegeneration can occur in all neural circuits, even those involved in simpler brain functions. For example, degeneration is found in the primary visual cortex as well as the hippocampal formation in most transgenic mouse models of degeneration. The visual cortex potentially provides a simpler and better-understood model for investigating the functional consequences of neurodegenerative diseases, because there are well established paradigms for studying intrinsic plasticity in the visual cortex, and its behavioural correlates. But which, if any, forms of visual cortical plasticity are impaired in mouse models of neurodegeneration is not known. Indeed, it is not yet clear that degeneration in visual cortex is accompanied by any functional changes. The characteristic orientation selectivity of neurons in V1 is preserved, even in late stages of degeneration, in mice that overexpress amyloid-β precursor protein (APP)[7,8], and in the widely used and well-characterised rTg4510 mouse model of tauopathy[9].

We measured visual plasticity in the rTg4510 mouse model[10,11]. The rTg4510 mice develop progressive neurofibrillary tangles (NFT), neuronal loss and concomitant cognitive deficits[10,12]. High levels of mutant tau emerge in the hippocampus and neocortex (including visual cortex) between 2 and 4 months of age, and NFT are present by 4.5 months in hippocampus and 7–8 months in neocortex[10,11]. Significant reduction in the size of cortical areas[13] and cortical cell loss (~52%) occurs by 8.5 months[14]. We therefore studied animals at early stages of tauopathy (~5 months old) where there is significant tau accumulation but no cortical cell loss, and at a more advanced stage where cortical degeneration has taken place (~8 months). To characterise intrinsic plasticity, we studied responses to repeated presentation of a simple visual pattern. Over several minutes, repeated presentation of a visual pattern usually suppresses visual cortical responses to that pattern, a classical effect of sensory adaptation[15,16]. Over several days, however, repeated presentation of a visual pattern can instead increase response, involving a sleep-dependent process called stimulus response potentiation, or SRP[17–19]. SRP is a form of long-term plasticity that resembles canonical LTP, with which it shares mechanisms[19].

We found that basic visual evoked responses were largely unaffected, even at late stages of tauopathy, consistent with previous reports[9]. However, both short- (intra-day) and long-term (inter-day) visual plasticity were disrupted, even at early stages of tauopathy. Both timescales of visual plasticity were further impaired in older animals, such that they were abolished at later stages of tauopathy. Additionally, we found that innate visually evoked behaviours were impaired in mice expressing the mutant tau. Our results therefore show that there are substantial changes in intrinsic visual cortical plasticity and in visual behaviour even early in tauopathy in rTg4510 mice. Visual plasticity may therefore provide a potential target for detecting dementias at early stages of the disease, in humans as well as in animal models.

## Results

We measured the local field potential (LFP) from the primary visual cortex (V1) of 50 rTg4510 mice, using electrodes targeted to layer 4. Half of the mice were placed on a doxycycline diet from 2 months of age, to suppress the expression of mutant tau (referred as Tau−)[11,13,14]. We used different animals to study early stages of tauopathy (~5 months) and more advanced stages (~8 months) (Fig. 1a). Mice fed a normal diet, and therefore continuing to express the mutant tau (referred as Tau+) showed increased tau pathology ($F = 23.61$, $p = 10^{-4}$, two-way ANOVA, Fig. 1b, Supplementary Fig. 1) and a significant reduction in the overall brain weight compared to Tau− animals ($F = 5.79$, $p = 0.02$, Fig. 1b).

**Visual cortical responses in rTg4510 mice**. We recorded the visual evoked potential (VEP) in response to a large sinusoidal grating, presented to the monocular visual field by a computer monitor. Mice were head-fixed but free to run on a styrofoam wheel. The contrast of the grating was flipped (reversed) every 0.5 s, and the animals were exposed to 10 blocks of 200 reversals, with 30 s of a grey screen presented between blocks. Each reversal generated a VEP, with an initial negative deflection rapidly followed by a positive one (Fig. 1c). We calculated the VEP amplitude as the difference between the positive and negative peaks of the VEP signal. We found that the amplitude of the VEP was similar in 5-month old Tau− and Tau+ animals (Fig. 1d). VEP amplitude was reduced for both 8-month old Tau− and Tau+ animals (Fig. 1e). We found a significant difference in the average amplitude between age groups ($F = 5.05$, $p = 0.03$, two-way ANOVA) but not between phenotypes ($F = 0.11$, $p = 0.75$). Post hoc comparison of 5 and 8 month old Tau+ mice showed slight reduction in VEP amplitude in older animals (Fig. 1e, $p = 0.04$, Tukey's test). We further calculated the amplitude of each of the positive and negative peaks, the time to those peaks, the full width at half maximum for those peaks, and the decay slope following the positive peak. We found a significant difference between age groups in the amplitude of the negative peak, and the time to negative and positive peaks, but no significant differences between phenotypes for any measure (Supplementary Table 1). The VEP signal was slightly more sustained in 8-month old animals, for both Tau− and Tau+ animals, but the effect did not reach significance (Supplementary Table 1). Our results, therefore, show that basic visual cortical responses are largely preserved in the rTg4510 mouse model, even during advanced tauopathy.

**Visual plasticity is disrupted in mice expressing mutant Tau**. We hypothesised that plasticity is more likely to be affected than the basic visual response at early stages of tauopathy. Stimulus-Response Potentiation (SRP) is a form of intrinsic cortical plasticity induced by repeated exposure to a visual stimulus over several days that can be measured in the VEP of mouse V1[17]. To measure SRP in these animals we obtained VEP measurements from 23 5-month old rTg4510 mice (12 Tau− and 11 Tau+) while they were exposed to a grating of one orientation (either 45° or −45° from vertical) over 9 days (thus becoming a familiar stimulus). On the first and last day of recordings (day 1 and day 9), five blocks of this familiar stimulus orientation were interleaved with five blocks of the orthogonal orientation (unfamiliar stimulus). On days 2–8, 10 blocks of the familiar stimulus were presented (Fig. 2a).

We found strong potentiation of the VEP signal in Tau− animals, and reduced and slower potentiation in Tau+ animals (Fig. 2). The VEP amplitude for Tau− mice was already increased on day 2, and reached a plateau around day 3–4 (Fig. 2b;

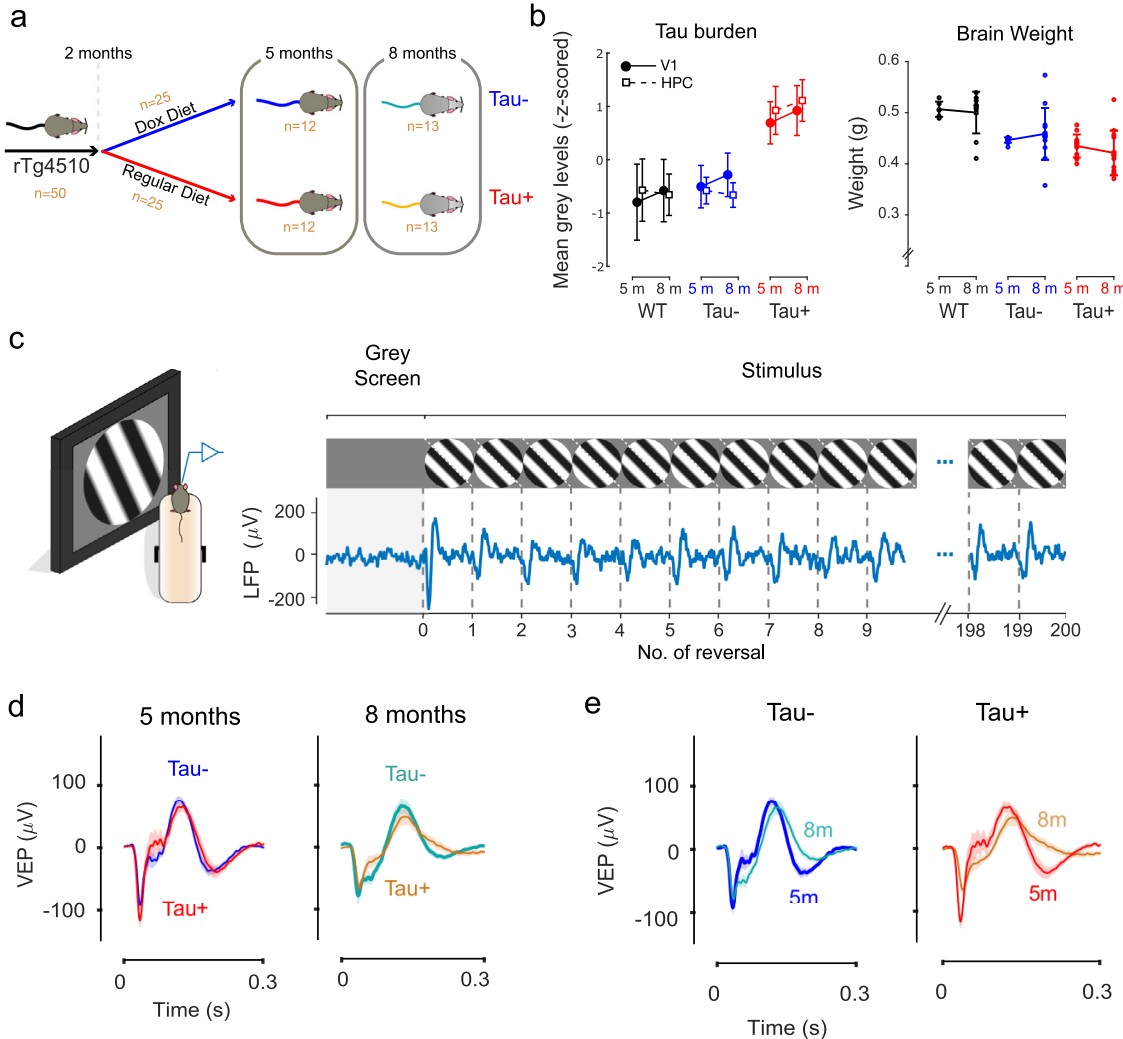

**Fig. 1 Visual evoked responses in the primary visual cortex are largely preserved in tauopathy. a** 50 rTg4510 mice were used in this study. At 2 months of age, half the rTg4510 mice ($n = 25$) were treated with and fed doxycycline (Dox) to suppress the expression of the transgene and arrest further accumulation of Tau (Tau−). 12 of those Tau− mice were studied at 5 months of age and the other 13 at 8 months. Of the 25 mice fed a normal diet (Tau+), 12 were studied at 5 months and the remaining 13 at 8 months. Data from one 5-month old Tau+ mouse was subsequently removed from all electrophysiological analysis because the electrode was found to be placed too deep in the cortex (see "Methods"). **b** Left, comparison of average tau burden for each group of animals calculated as the mean grey levels within selected regions of interest in V1 and hippocampus (HPC) ("Methods"). Data are represented as mean ± 2*SEM. Right, comparison of overall brain weights at 5 months (5 m) and 8 months (8 m) timepoints. Data are represented as mean ± SEM. **c** Mice were head-fixed and allowed to run on a styrofoam wheel. After 5 min exposure to a homogenous grey screen, a full-screen, contrast reversing, sinusoidal grating pattern was presented to the left monocular visual field. The grating reversed every 0.5 s, and the animal was exposed to 10 blocks of 200 continuous reversals, with 30 s presentation of a grey screen between blocks. We recorded the local field potential (LFP) in the right primary visual cortex (V1). The LFP trace shown is the average time course across 10 stimulus blocks on one day in one Tau− mouse. Each reversal generated a visual evoked potential (VEP), characterised by an initial negative deflection and subsequent positive one. **d** Average VEP responses on the first day of recording for Tau− and Tau+ animals at 5 months (left) and 8 months (right). There was no significant difference in the size and shape of the VEPs between groups at either age ($p = 0.75$, two-way ANOVA). **e** Average VEP responses for Tau− (left) and Tau+ (right) mice obtained at 5 months and 8 months of age. VEPs were slightly reduced, and more sustained, for both Tau− and Tau+ mice at 8 months. Shaded area represents the mean±SEM (5-month-old: $n$=12 Tau−, $n = 11$ Tau+; 8-month-old: $n = 13$ Tau−, $n = 13$ Tau+). The source data underlying this figure are available in Supplementary Data 1.

comparison to day 1, for days 2: $p = 0.001$, 3: $p = 1.2*10^{-4}$, 4: $p = 1.2*10^{-5}$, 5: $p = 1.9*10^{-4}$, 6: $p = 2.2*10^{-5}$, 7: $p = 2.8*10^{-6}$, 8: $p = 9.7*10^{-5}$, 9: $p = 3.8*10^{-5}$, repeated measures ANOVA, Tukey's pairwise comparisons). Tau+ mice showed a slower increase in the VEP amplitude, showing a significant increase on day 4 compared to day 1, and reaching a plateau around day 5–6 (comparison to day 1 for days 2: $p = 0.452$, 3: $p = 0.354$, 4: $p = 0.005$, 5: $p = 0.014$, 6: $p = 1.9*10^{-4}$, 7: $p = 2*10^{-4}$, 8: $p = 0.001$, 9: $p = 2.4*10^{-4}$). For example, the VEP amplitude of Tau− mice was $77 \pm 11$ µV larger on day 3 than it was on day 1

(Fig. 2b, c), while Tau+ animals showed only a moderate increase by day 3 ($31 \pm 15$ µV). By day 9, VEP amplitude had increased substantially for both groups of mice (Fig. 2d). Repeated measures ANOVA revealed a significant effect of day ($F = 23.7$, $p = 4.1*10^{-24}$) but no day-by-phenotype interaction ($F = 1.11$, $p = 0.35$). Post hoc comparisons between groups on the change in VEP amplitude (difference between the VEP amplitude on each day and VEP amplitude on day 1; Fig. 2b) showed a significant difference for day 3 (day 2: $p = 0.08$, day 3: $p = 0.019$, day 4: $p = 0.088$, day 5: $p = 0.298$, day 6: $p = 0.62$, day 7: $p = 0.23$, day 8:

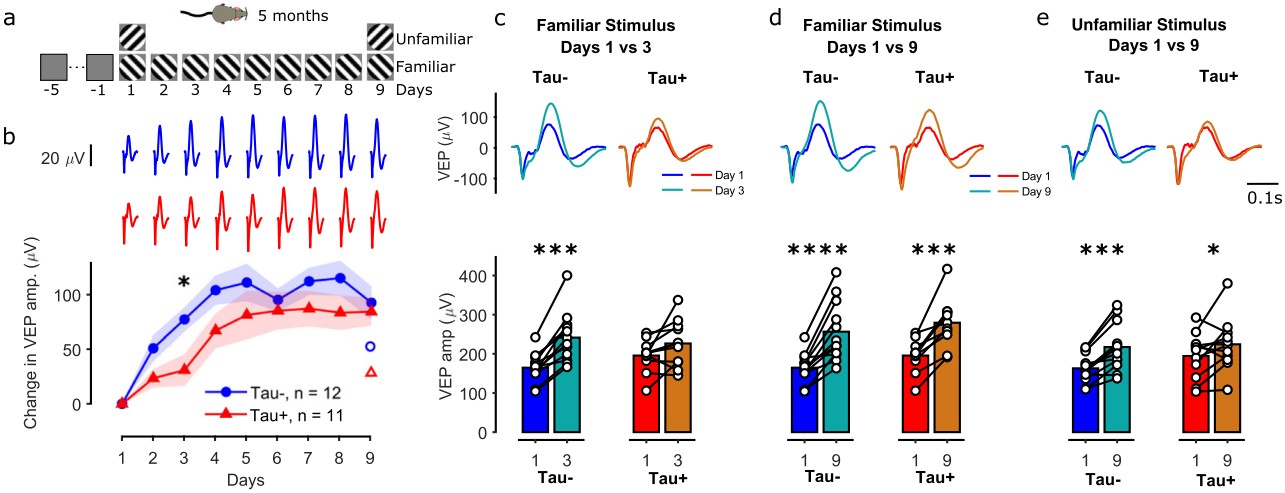

**Fig. 2 Visual plasticity is reduced at early stages of tauopathy. a** Mice (5 months old) were exposed to a grating of one orientation (either 45° or −45°; 'familiar' stimulus) for 9 days. On the first and last day of recordings, five blocks of this familiar stimulus were interleaved with five blocks of a grating ('unfamiliar') whose orientation was orthogonal to the familiar grating. On days 2–8, 10 blocks of the familiar stimulus were presented. **b** Change in the VEP amplitude from day 1 (VEP amplitude on each day, minus the VEP amplitude on day 1, calculated separately for each animal before averaging) for the familiar stimulus, for Tau− (blue) and Tau+ (red) animals. Tau+ mice showed a slower potentiation of the LFP signal compared to Tau− animals. Shaded area represents the mean ± SEM ($n = 12$ Tau−, $n = 11$ Tau+). The asterisk denotes significance between the two groups on the respective day. The open symbols on the right show the change in VEP amplitude in response to the unfamiliar stimulus on day 9, compared to day 1. **c–e** Comparison of average VEPs (top) and VEP amplitudes (bottom) between: days 1 vs 3 for the familiar stimulus (**c**), days 1 vs 9 for the familiar stimulus (**d**), and days 1 vs 9 for the unfamiliar stimulus (**e**). ****$p < 10^{-4}$, ***$p < 10^{-3}$, *$p < 0.05$. The source data underlying this figure are available in Supplementary Data 2.

$p = 0.176$, day 9: $p = 0.686$, Tukey's). The slower growth in VEP amplitude in Tau+ animals was mainly due to slower changes in the positive deflection of the VEP signal (Supplementary Fig. 2b). Repeated measures ANOVA for the positive peak showed a significant day-by-phenotype interaction ($F = 2.13$, $p = 0.035$). Post hoc comparisons on the change in VEP amplitude of the positive peak showed a significant difference between Tau+ and Tau− mice for days 2–5 (day 2: $p = 0.028$, day 3: $p = 0.003$, day 4: $p = 0.02$, day 5: $p = 0.017$, day 6: $p = 0.31$, day 7: $p = 0.129$, day 8: $p = 0.097$, day 9: $p = 0.38$, Tukey's). The negative deflection of the VEP was slightly larger in Tau+ mice than in Tau− mice, and increased at a similar rate in both groups (Supplementary Fig. 2c; day: $F = 6$, $p = 9.9*10^{-7}$, day-by-phenotype: $F = 0.6$, $p = 0.8$). A significant increase over days was also observed for the time to negative peak ($F = 4.3$, $p = 9*10^{-5}$), time to positive peak ($F = 6.1$, $p = 6.7*10^{-7}$) and the width at half maximum of the positive peak ($F = 8.3$, $p = 1.8*10^{-9}$), but there was no significant day-by-phenotype interaction (time to positive peak: $F = 0.26$, $p = 0.97$, time to negative peak: $F = 0.85$, $p = 0.56$, width of positive peak: $F = 0.54$, $p = 0.82$).

Repeated exposure to the familiar stimulus had less impact on VEP response to the unfamiliar stimulus (which was shown only on day 1 and day 9), for both groups (Fig. 2e, Tau−: unfamiliar VEP amp. change $= 53 \pm 11$, $p = 2.9*10^{-4}$, Tau+: $29 \pm 15$, $p = 0.03$, Tukey's test). Repeated measures ANOVA showed a significant effect of day ($F = 21.6$, $p = 1.3*10^{-4}$) but no day-by-phenotype interaction ($F = 1.82$, $p = 0.19$). We saw no difference between phenotypes in the amplitude change from day 1 to day 9 ($p = 0.2$). In fact, growth in VEP responses to the unfamiliar stimulus on day 9 was comparable to growth in VEPs to the familiar stimulus on day 2 (Fig. 2b, Tau−: day 2 familiar VEP amp. change $= 51 \pm 12$, $p = 0.89$, Tau+: $23 \pm 8$, $p = 0.76$, t-test on the VEP amp. change between familiar stimulus on day 2 and unfamiliar stimulus on day 9). Together these data suggest that the amplitude but not stimulus selectivity of response potentiation is affected in Tau+ animals.

Overall our results suggest that visual cortical plasticity is disrupted even at this early stage of tauopathy.

**Visual plasticity is reduced in older animals.** We showed that visual plasticity is affected even at early stages of tauopathy in rTg4510 mice. We then asked whether functional deficits increase with age by examining visual cortical plasticity in 8-month old rTg4510 mice. We obtained LFP responses from 26 8-month old transgenic mice (13 Tau− and 13 Tau+) using the same visual paradigm described above (Fig. 3a).

Visual plasticity was reduced in older animals (Fig. 3). By day 9, the VEPs of Tau− mice showed a small increase relative to day 1 (Fig. 3c, VEP amp. change $= 39 \pm 20 \mu V$), while Tau+ animals showed minimal change ($13 \pm 9 \mu V$). Repeated measures ANOVA including all days showed no effect of day (Amp.: $F = 1.65$, $p = 0.11$, positive amp.: $F = 1.73$, $p = 0.093$, negative amp.: $F = 1.42$, $p = 0.189$) or interaction effects (Amp: $F = 0.7$, $p = 0.68$, positive amp.: $F = 0.64$, $p = 0.742$, negative amp.: $F = 1.46$, $p = 0.173$) for any VEP property measured. If only days 1 and 9 were considered, repeated measures ANOVA showed a significant effect of day ($F = 5.53$, $p = 0.027$) and post hoc comparisons showed a significant difference between day 1 and day 9, for Tau− (0.019) but not Tau+ mice ($p = 0.414$, Tukey's). Post hoc comparisons on the change in VEP amplitude (Fig. 3b) showed no significant difference between groups for either day (day 2: $p = 0.88$, day 3: $p = 0.37$, day 4: $p = 0.44$, day 5: $p = 0.83$, day 6: $p = 0.54$, day 7: $p = 0.68$, day 8: $p = 0.31$, day 9: $p = 0.25$, Tukey's). No significant change was observed for the unfamiliar stimulus in either group (Fig. 3d, Tau−: VEP amp. change $= 25 \pm 16$, $p = 0.72$, Tau+: $6 \pm 9$, $p = 0.63$). Our results suggest that there may be a combined effect of mutant tau expression and age leading to a complete disruption of visual cortical plasticity.

**Similar response potentiation in Tau− and WT mice.** Tau− mice were fed with a doxycycline diet to suppress mutant tau transgene expression from the age of 2 months. It is possible however that expression of mutant tau up to the age of 2 months, or a continued effect thereafter, could lead to functional deficits in these animals. For example, the reduced visual plasticity observed

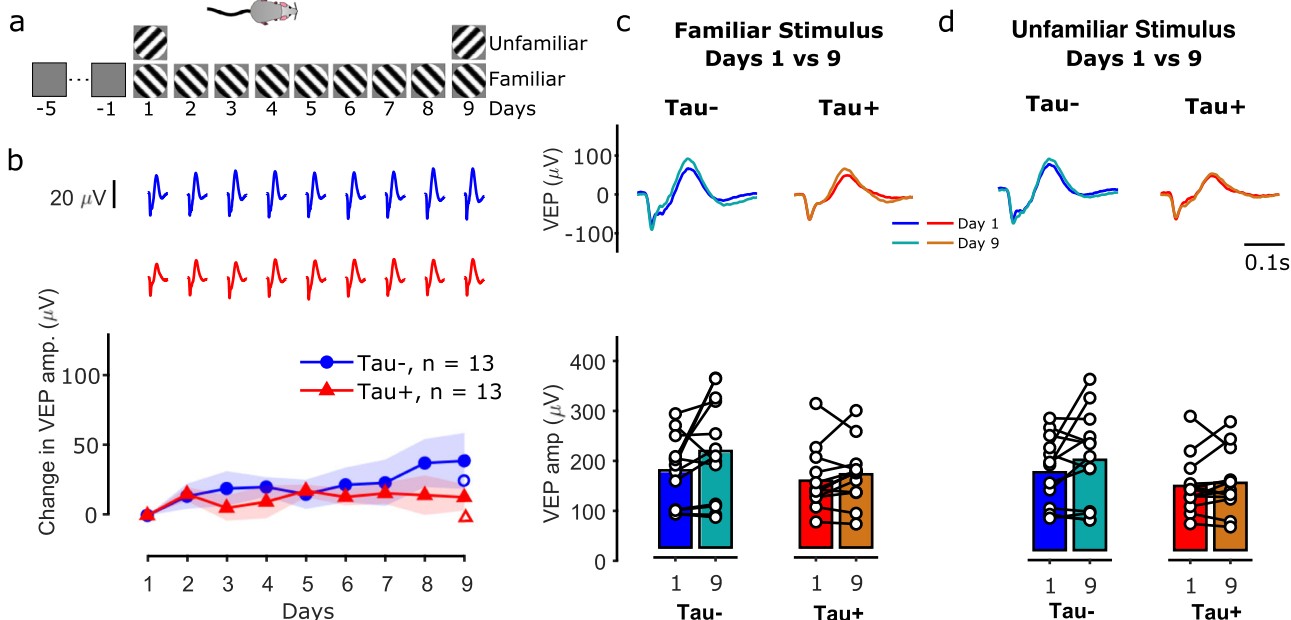

**Fig. 3 Visual plasticity diminishes with age. a** Mice (8 months old) were exposed to a grating of one orientation (either 45° or −45°; 'familiar' stimulus) for 9 days (same as in Fig. 2a). **b** Difference in the VEP amplitude from day 1 for the familiar stimulus for Tau− (blue) and Tau+ (red) animals over the course of days. 8-month old Tau− mice showed a smaller potentiation of the LFP signal compared to 5-month old mice (Fig. 2b). Tau+ mice showed no potentiation. Shaded area represents the mean±SEM (n = 13 Tau−, n = 13 Tau+). The open symbols on the right show the change in VEP amplitude in response to the unfamiliar stimulus on day 9. **c, d** Comparison of average VEPs (top) and VEP amplitudes (bottom) between: days 1 vs 9 for the familiar stimulus (**c**), and days 1 vs 9 for the unfamiliar stimulus (**d**). The source data underlying this figure are available in Supplementary Data 3.

in 8-month old compared to 5-month old Tau− mice might be a pathological effect initiated by early tauopathy. We, therefore, compared the visual responses of Tau− mice with responses obtained from wild type (WT) littermates.

At 5 months, WT mice showed similar VEP potentiation to the Tau− mice (Supplementary Fig. 3a). By day 3, WTs showed an increase in the VEP amplitude for the familiar stimulus relative to day 1 that was similar to Tau− mice (Supplementary Fig. 3a, WT: VEP amp. change = 50 ± 14 μV, Tau−: 77 ± 11 μV, $p = 0.167$, Tukey's). Repeated measures ANOVA showed no day by phenotype interaction for either the VEP amplitude ($F = 1.28$, $p = 0.258$) or the positive ($F = 1.28$, $p = 0.255$) and negative peak amplitudes ($F = 0.94$, $p = 0.485$). At 8 months, the VEP amplitude was on average larger in WT compared to Tau− mice ($F = 0.92$, $p = 0.35$, one-way ANOVA). As for Tau− mice, however, 8-month old WT mice showed reduced and slower potentiation of the LFP signal for the familiar stimulus compared to 5-month old mice (Supplementary Fig. 3b). This suggests that the reduction in SRP at 8 months is largely an effect of age and not tauopathy in Tau− mice.

**Coincident changes in short-term visual plasticity**. We have shown that long-term visual plasticity is disrupted in rTg4510 mice using a simple and well-established visual paradigm. Although SRP has been conventionally used as a measure of plasticity across days, it can also be used to measure changes within a day, or within a block (often called adaptation). We, therefore, asked if there were also disruptions in plasticity at these shorter timescales. We analysed how responses changed within each day. We considered days 2–8 where only the familiar stimulus was presented (Fig. 4a). Responses gradually increased over the course of the first block of visual stimuli in all animals. In subsequent blocks, the amplitude of the VEPs usually reduced over the course of each block, consistent with classic sensory adaptation effects (Fig. 4b). We, therefore, focussed our analyses

on the second and subsequent stimulation blocks. To quantify adaptation's effects, we fitted a decaying exponential function to the within-block time course of the VEP amplitude. We obtained the within-block time-course by subtracting the steady-state VEP amplitude (the average of the last 100 reversals) from each block, and then averaging across blocks. We first averaged VEP responses across blocks 2–10 and found smaller adaptation effects in Tau+ than Tau− mice (Fig. 4c left panel; Tau−: 42 ± 31 μV, Tau+: 28 ± 44, $p = 0.0215$, t-test). Adaptation effects were reduced in 8-month old Tau− mice compared to 5-month olds, and they were completely abolished in 8-month Tau+ mice (Fig. 4e, f; Tau−: 15 ± 22, Tau+: 0 ± 24, $p < 0.0001$).

The disruption to adaptation effects in 5-month old Tau+ mice was more apparent in later blocks than early blocks (Fig. 4c). We calculated average VEP amplitudes separately for early blocks (2–5) and late blocks (6–10), and estimated adaptation's effect for both sets for each recording day in each animal (Fig. 4d). Tau+ animals showed less adaptation effects than Tau− mice in blocks 6–10 (Tau−: 47 ± 44 μV, Tau+: 26 ± 60, $p = 0.0131$, t-test), but similar adaptation effects in blocks 2–5 (Tau−: 36 ± 43 μV, Tau+: 34 ± 58 $p = 0.7414$, t-test). Within-block adaptation effects were absent in 8-month Tau+ mice in both early and late blocks (Fig. 4f, g; blocks 2–5: Tau−: 14 ± 29, Tau+: −9 ± 39, $p < 0.0001$; blocks 6–10: Tau−: 16 ± 25, Tau+: 6 ± 28, $p = 0.0131$, t-test). We conclude that adaptation effects are disrupted early in tauopathy, and abolished at later stages.

We found that unlike Tau− animals, 8-month old WTs showed similar within block adaptation effects to 5-month old WTs, suggesting that while long term plasticity is reduced in older WT animals, short term plasticity is not (Supplementary Fig. 4, Fig. 4; 5 m WT: 26 ± 29; 8 m WT: 27 ± 31). Reduced adaptation in older Tau− animals compared to WT may reflect persistent impact of the initial mutant tau expression before the onset of the doxycycline treatment (which was started at 2 months), subsequent incomplete suppression of the transgene, or age-dependent effects of other genetic disruptions in this mouse model (see "Discussion").

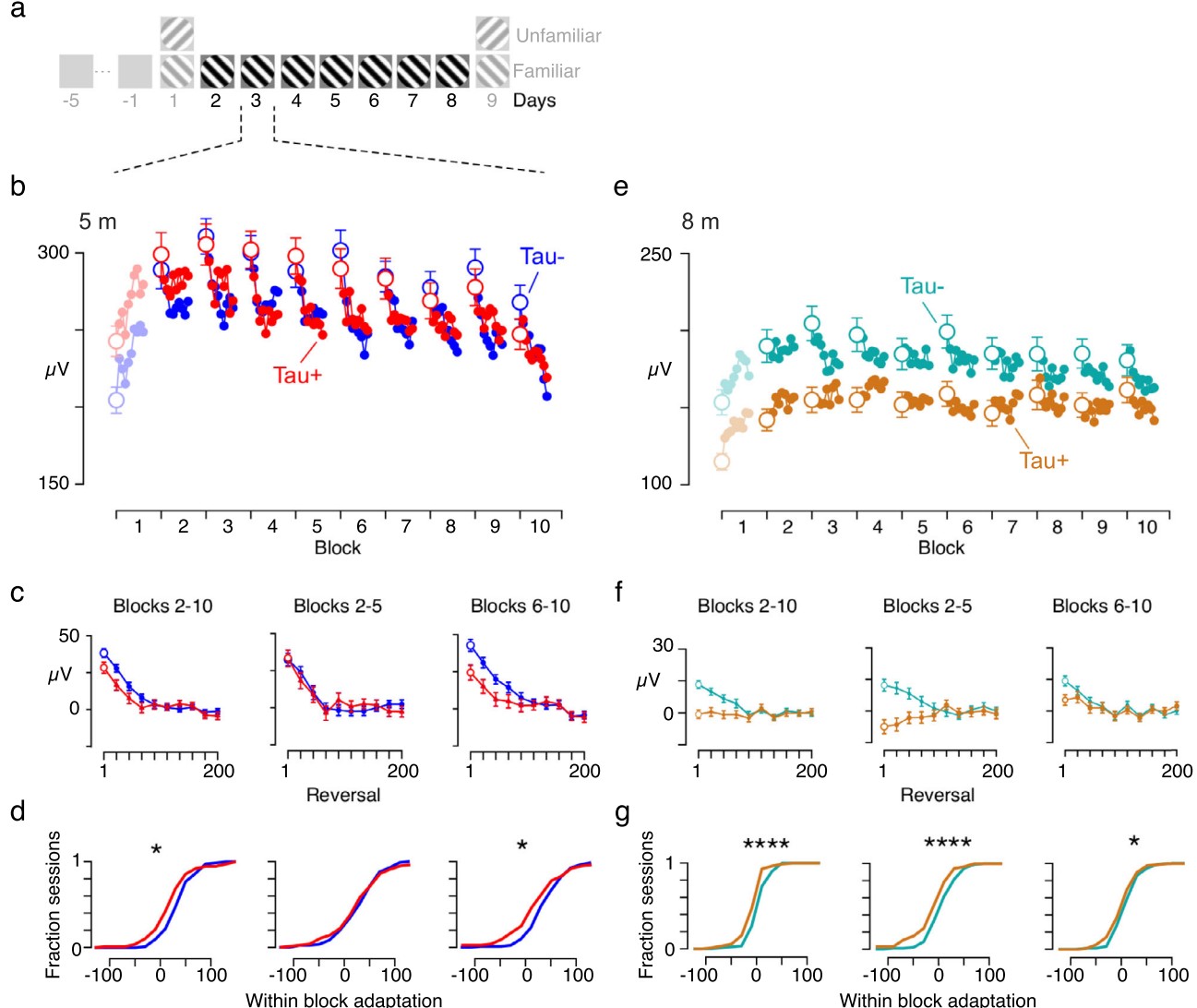

**Fig. 4 Short-term visual plasticity is reduced in rTg4510 mice expressing mutant tau. a** We measured intra-day effects on experimental days 2–8. On each day 10 blocks of the familiar stimulus were presented, each consisting of 200 phase reversals and separated by 30 s of a homogenous grey screen. **b** Average VEP amplitude for days 2–8 as a function of block number for 5-month old Tau− (blue, $n = 12$) and Tau+ (red, $n = 11$) mice. The VEP amplitude was calculated from non-overlapping averages of 20 reversals. With the notable exception of the first block, VEPs showed a reduction of responses within each block, consistent with classic sensory adaptation effects. **c** Average VEP responses for blocks 2–10 (Left) and separately for blocks 2–5 (Middle) and blocks 6–10 (Right) on days 2–8, for Tau− ($n = 12$) and Tau+ ($n = 11$) animals. VEP amplitudes within each block were normalised by subtracting the average VEP amplitude of the last 100 reversals for that block. The within block amplitudes (at a resolution of 20 reversals) were then fit with a decaying exponential function of fixed $\tau = 8.4$ reversals (see "Methods"), and the amplitude of the exponential was extracted. **d** Cumulative histograms of these fitted amplitudes for Tau+ (solid red) and Tau− (dotted blue) mice, normalised by the total number of sessions for each cohort. At 5 months, Tau+ animals showed reduced within-block adaptation compared to Tau− mice for blocks 6–10, but not for blocks 2–5. The asterisks indicate a significant difference between the distributions (Students t-test; ****$p < 10^{-4}$, *$p < 0.05$). **e–g** Same as **b–d** for 8-month old animals ($n = 13$ Tau−, $n = 13$ Tau+). The within-block adaptation is reduced for Tau− mice at 8 months compared to 5 months. Tau+ mice showed no suppressive adaptation effect for either early or late blocks. Error bars represent the mean ± SEM. The source data underlying this figure are available in Supplementary Data 4.

**Differences in visual plasticity cannot be explained by differences in behavioural state.** Animals were free to run on a styrofoam wheel during the recording session, and we observed epochs of running and variations in pupil size in all animals. Visual cortical responses, as well as responses earlier in the visual pathway, are known to vary with behavioural state as defined by running speed or pupil area[20]. Locomotion has also been shown to enhance stimulus-specific plasticity in the adult visual cortex[21]. Tau+ rTg4510 mice have been reported to have abnormal locomotion behaviours[13,22,23]. We, therefore, wanted to know if the differences observed in visual plasticity between Tau+ and Tau− animals might be explained by differences in behavioural state.

At 5 months Tau− mice were slightly more active during stimulus presentation compared to Tau+ mice, spending on average $17 \pm 16.8\%$ (mean±standard deviation) of their time running compared to $10 \pm 8.7\%$ for Tau+ mice ($p = 2.7*10^{-4}$, t-test). The average amount of time spent running did not change across days. Repeated measures ANOVA showed no effect of day on the average speed ($F = 0.73$, $p = 0.66$) or day-by-phenotype interaction ($F = 1.63$, $p = 0.12$). We first asked whether the amount of running predicted the amount of plasticity. We found no correlation between the amount of time animals spent running during stimulus presentation on one day, and the amount of SRP on the next day ($r = 0.07$, $p = 0.29$). We also found no correlation

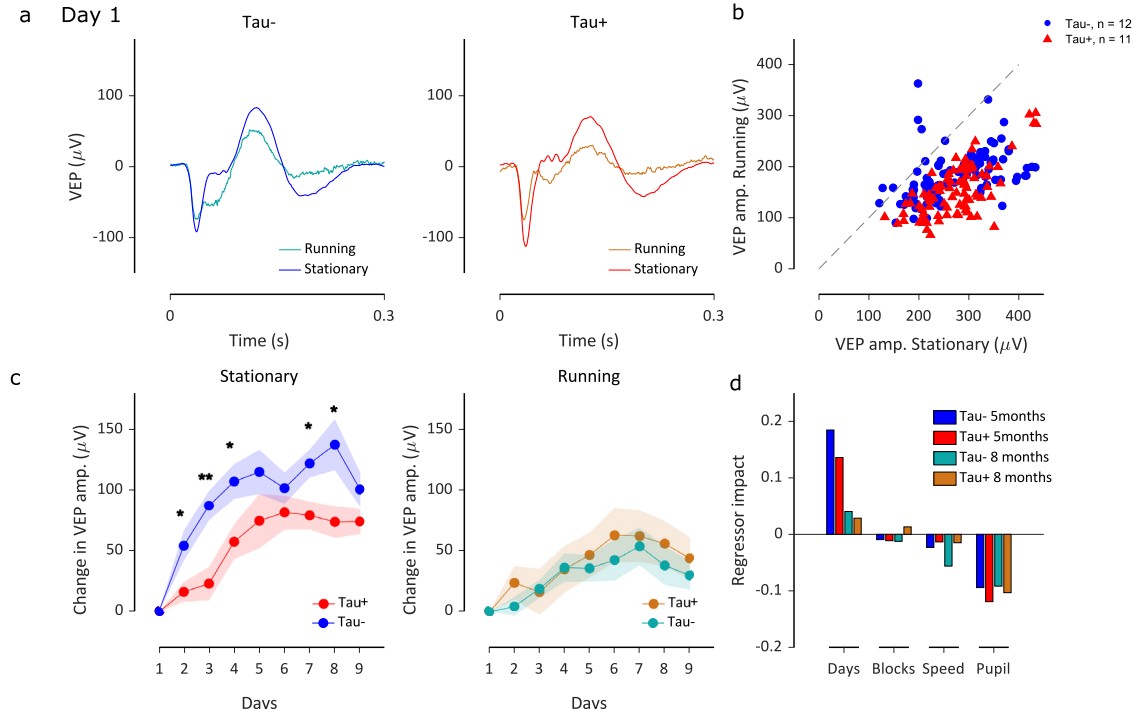

**Fig. 5 Differences in visual plasticity are not due to differences in behavioural state. a** VEP responses on day 1 for Tau− (blue, $n = 12$) and Tau+ (red, $n = 11$) 5-month old animals calculated during running (speed > 1 cm/s) and stationary epochs. Running reduces the VEP signal for both groups of mice. **b** VEP amplitude during stationary versus running epochs for individual Tau− (blue circles) and Tau+ (red triangles) mice. Each symbol represents the VEP amplitude on one day in one animal. **c** Difference in the VEP amplitude from day 1 for the familiar stimulus, for Tau− and Tau+ 5-month old animals, calculated for only stationary (left) or only running (right) epochs. VEP potentiation was evident in both groups of animals for both behavioural states. Shaded area represents the mean ± SEM ($n = 12$ Tau−, $n = 11$ Tau+). **p < 0.01, *p < 0.05. **d** We fitted an elastic net regularisation model to predict the VEP amplitude for each animal using days, block number, movement speed, and pupil diameter as regressors. We assessed the impact of each regressor in predicting the VEP amplitude as the product between each learned weight and observed variable, divided by the predicted value across the entire regularisation path (of elastic net regression). The panel shows the average regressor impact for each phenotype at 5 and 8 months of age. The source data underlying this figure are available in Supplementary Data 5.

between the average activity of animals and the total amount of SRP found on day 9 ($r = −0.26$, $p = 0.16$). Therefore, running behaviour could not predict visual plasticity in 5 month old animals. At 8 months, Tau− mice were more active (time spend running = $33 ± 31\%$) than Tau+ mice ($13 ± 10\%$; $p = 3.2*10^{-10}$). There was a small positive correlation between the amount of time animals spent running during stimulus presentation on one day and the amount of SRP on the next day ($r = 0.16$, $p = 0.006$). However, visual plasticity was significantly reduced at 8 months for both groups.

We then asked how the behavioural state of the animal affects visual responses and whether this can explain differences in plasticity between Tau− and Tau+ mice. We used two approaches. We first refined our analyses by only considering VEP responses during either stationary (speed < 1 cm/s) or running (speed > 1 cm/s) epochs. We next used a model to evaluate the contribution of various parameters to the observed responses.

The VEP signal was smaller during running, in both Tau− and Tau+ mice (Fig. 5a). This reduction was seen across all days and animals (Fig. 5b), and was slightly more pronounced in Tau+ animals. To establish whether the animal's locomotion state was responsible for the differences observed in SRP between groups, we calculated the change in the VEP amplitude from day 1, but considering only stationary or only running epochs (Fig. 5c; Supplementary Fig. 5b). The differences in SRP between 5-month-old Tau− and Tau+ mice were more pronounced when we considered only stationary epochs (Fig. 5c and Supplementary

Fig. 5a–c). Repeated measures ANOVA showed a significant day-by-phenotype interaction ($F = 2.07$, $p = 0.041$) for the VEP amplitude. Post hoc comparisons between groups on the change in VEP amplitude (Fig. 5c) showed a significant difference for days 2,3,4,7,8 (day 2: $p = 0.02$, day 3: $p = 0.002$, day 4: $p = 0.022$, day 5: $p = 0.171$, day 6: $p = 0.298$, day 7: $p = 0.015$, day 8: $p = 0.019$, day 9: $p = 0.146$, Tukey's). The difference in SRP was attenuated when only running epochs were considered (Fig. 5c; repeated measures ANOVA, day: $F = 7.17$, $p = 4.2*10^{-8}$, day-by-phenotype: $F = 0.39$, $p = 0.92$). 8-month-old animals showed reduced plasticity which did not significantly increase when only stationary epochs were considered (Supplementary Fig. 5b–d). Therefore, we conclude that differences in running behaviour cannot explain the differences in SRP between Tau− and Tau+ mice.

Our analyses show that VEP amplitude is dependent on behavioural state, and varies within a day (Fig. 4). These effects might combine to accentuate or mask plasticity across days. To assess the relative impact of time and behaviour on the VEP, we used an elastic net regularisation model to predict the VEP amplitude for each animal using *Day*, *Block* number, movement *Speed*, and *Pupil* diameter as regressors. The model assigns weights to the defined features (regressors) in order to predict the target variable (VEP amplitude). We explored the model solutions for a range of 10000 values of the elastic net's regularisation hyperparameters ("Methods"). We then estimated the impact of each regressor, based on the derived regression weights, as the percentage of contribution to the estimates of the

VEP amplitude across the entire regularisation path. We found *Day* to have a positive impact on the VEP amplitude, thus predicting the increase in VEP amplitude across days (Fig. 5d). Consistent with our observations on visual plasticity (Figs. 2, 3), the impact of *Day* in predicting the VEP amplitude was greater for Tau− than Tau+ animals, for both 5-month old and 8-month old animals. Similarly, *Day* had a lower impact at 8 months than 5 months old. *Block* number had a negligible overall impact in predicting the VEP amplitude compared to the other predictors considered here. Increases in behaviour (defined by pupil diameter and speed) had a negative impact on the VEP amplitude, and this relationship was similar across phenotypes and age groups. In addition, there was no correlation between behaviour and days (Pupil: $r = -0.03$, Speed: $r = 0.07$) suggesting that behaviour cannot explain the VEP increase across days. Therefore, our results suggest that the VEP potentiation to the familiar stimulus across days is an effect of experience to the visual stimulus and not an effect of changes in the behavioural state of the animals.

**Visual evoked behaviours are reduced in naive Tau+ mice.** In animals, including mice, an unexpected or unfamiliar visual stimulus usually evokes instinctive behavioural responses. In head-fixed animals who are unable to run, these behavioural responses can include low amplitude muscle movements ('fidgeting'). SRP in the mouse visual cortex has previously been associated with habituation of fidgeting[19]. To establish whether reduced plasticity in Tau+ mice is associated with reduced behavioural responses in our conditions, we assessed the impact of stimulus presentation on pupil diameter and movement speed, both at first exposure to the stimulus, and later when animals were more experienced.

Tau− (and WT) mice showed large dilation of the pupil and an increase in the movement speed at the onset of their very first exposure to the grating stimulus (day 1, block 1; Fig. 6a, b and Supplementary Fig. 3). These responses quickly habituated, both within the session (Supplementary Fig. 6) and over the course of days (Fig. 6c, d). By contrast, Tau+ mice showed no visual evoked behavioural responses to the onset of the stimulus (Fig. 6a, b). These group differences in visual evoked behaviours were clear at both 5 and 8 months of age.

Experienced Tau− animals (days 6–8) showed a small constriction of the pupil in response to the stimulus onset. This constriction was more pronounced in Tau+ animals (Fig. 6c). There was no change in the movement speed for both groups in response to the stimulus, although experienced Tau− mice showed larger average speed compared with the Tau+ mice (Fig. 6d). Overall, our results suggest that visual evoked responses are disrupted even in early stages of tauopathy.

**Discussion**
In this study, we evaluated visual cortical plasticity in the rTg4510 mouse model of tauopathy. We measured both short term suppression and long-term potentiation of the visual evoked LFP response in V1, in mice with mutant tau expression (Tau+), or with that expression suppressed (Tau−). We made these measurements at two time points, at an early (5 months), and a more advanced (8 months) stage of tauopathy[10,11,14,24], which allowed us to estimate the progression of the pathology and its correlates in cortical plasticity. The results indicate that visual evoked responses are robust in both Tau− and Tau+ mice in both age groups. However, Tau+ animals show impaired visual cortical plasticity both within and across days. At 8 months of age, visual plasticity is reduced in Tau− animals, and practically abolished in Tau+ animals, potentially indicating a combined effect of age and tauopathy. In addition, we found an absence of behavioural

responses to novel visual stimuli in Tau+ animals. Overall, these data indicate that tauopathy has an impact on both long and short-term visual plasticity, and their potential behavioural correlates.

We have shown that naive visual evoked responses in V1 of rTg4510 mice are largely unaffected in Tau+ animals, even at advanced stages of tauopathy (8 months). These observations are consistent with previous work which shows robust orientation and direction selectivity in V1 of APP and rTg4510 mouse models[7–9]. The limited impact of degeneration on these basic visual properties of cortical neurons may suggest that the accumulation of tau is not sufficient to disrupt neuronal function[9]. However, orientation and direction selectivity may rely primarily on the functional properties of thalamocortical relay cells, and the pattern of their cortical projections, rather than intracortical operations[25]. These basic functional properties may therefore be resilient, because the thalamus is largely unaffected in many models of neurodegeneration, including the rTg4510 model. Functional properties that depend on cortical cellular mechanisms, and the balance of intra-cortical excitation and inhibition are more likely to be affected earlier in tauopathy[26]. Our observations support this hypothesis because visual cortical plasticity, which is likely to be more dependent on these processes, is disrupted in the rTg4510 mice.

We monitored visual plasticity at short- and long timescales. Stimulus response potentiation (SRP) is a form of long-term plasticity that is dependent on parvalbumin-positive interneuron activity[27], and is thought to co-opt similar pathways to thalamocortical LTP[28], including synaptic plasticity, NMDA receptor activation, and increased AMPA receptor trafficking[17,19]. Some of these circuits are also thought to be important in ocular dominance plasticity[29,30], so our observation of reduced SRP in Tau+ animals may be consistent with disruption of ocular dominance plasticity in the visual cortex of APP and PS1 mouse models[31,32].

Our experimental design allows for measurements at multiple timescales, and we found concomitant changes at short and long-time scales of plasticity. Short-term visual plasticity, usually known as adaptation, is linked to transient changes in the responsivity of synapses[33] or post-synaptic mechanisms related to spike frequency adaptation[34]. In 8-month old Tau+ animals, we found no sign of suppressive adaptation effects, even though the amplitude of the VEP was similar to that in Tau− animals, where we saw robust adaptation effects (albeit less than in 5-month old Tau− animals). Reduced adaptation effects in older Tau+ animals may reflect disruption of adaptation mechanisms in excitatory synapses or spike-frequency adaptation. There is also some evidence that adaptation effects include increased inhibition[35], so the absence of adaptation effects in older Tau+ animals may also reflect substantial disruption to inhibitory circuits. In 5-month old animals, adaptation effects in early blocks each day were similar in Tau+ and Tau− animals. In later blocks, adaptation effects were reduced in Tau+ animals. This reduced adaptation might reflect partial disruption to the same mechanisms that are then grossly impaired at 8-months, such that disruption to them is only revealed after prolonged bouts of stimulation. The reduced adaptation effects seen in later blocks may, however, also arise if additional suppressive adaptation effects in Tau+ animals are recruited over longer time-scales (that may exert an effect across blocks), or if mechanisms that allow recovery from the preceding block are impaired. Our results, therefore, suggest that even early stages of tauopathy have an impact on both synaptic mechanisms and intracellular trafficking in V1. We note, however, that because we measured the LFP, our measurements reflect the pooled signal of excitatory and inhibitory synapses[36,37]. Spiking activity in individual neurons, which depends on idiosyncratic and finely balanced excitation and inhibition, may show variable effects.

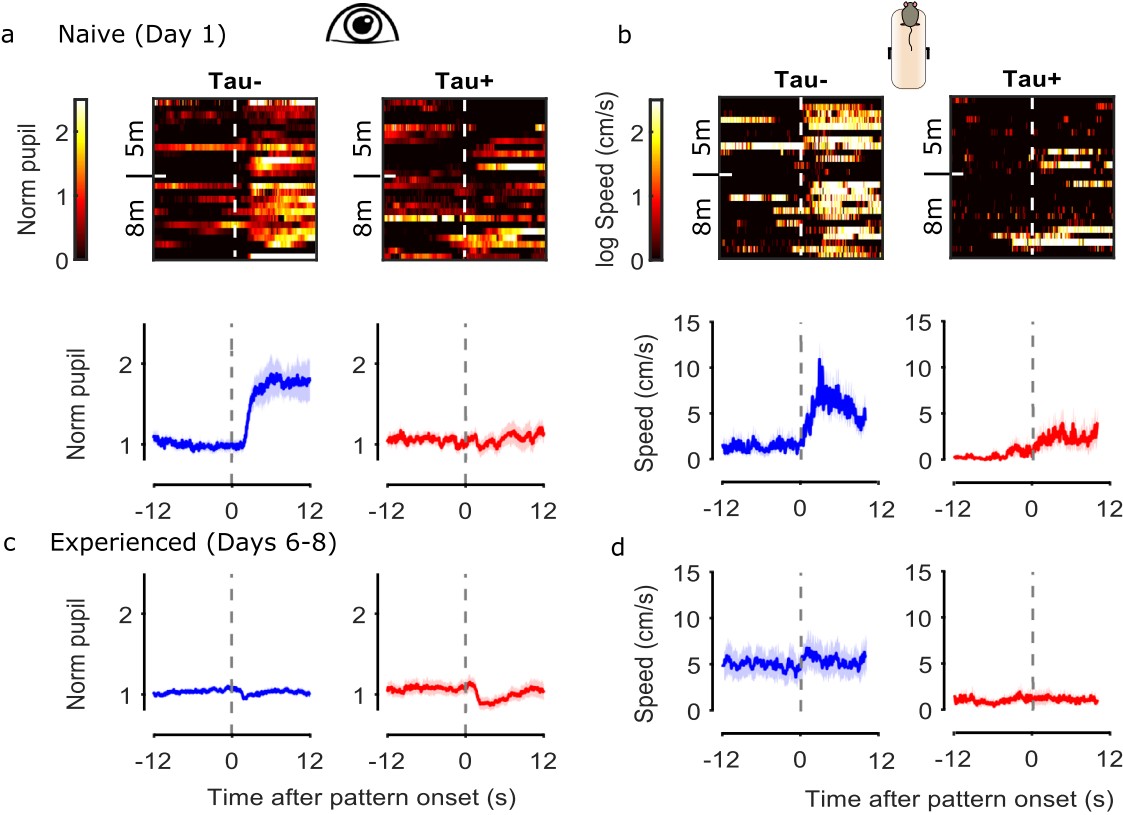

**Fig. 6 Visual evoked behaviours are reduced in Tau+ mice. a** *Top:* Images of the normalised pupil responses to the onset of the stimulus of the first block presented for naive animals (day 1; one animal per row). Pupil responses are normalised to the average pupil size during stationary epochs for 12 s before the stimulus onset where mice were viewing a grey screen. *Bottom:* Mean (±SEM) pupil responses for naive Tau− (blue, n = 25) and Tau+ (red, n = 25) animals for both age groups. **b** *Top:* Images of the natural logarithm of movement speed responses to the onset of the stimulus for naive animals (day 1). *Bottom:* Mean (±SEM) movement speed for naive Tau− (blue, n = 25) and Tau+ (red, n = 25) animals for both age groups. **c, d** Average normalised pupil responses and average movement speed to the onset of the stimulus for experienced animals (days 6–8). Horizontal dashed lines indicate the stimulus onset. Shaded areas represent the mean±SEM. The source data underlying this figure are available in Supplementary Data 6.

We have shown that a simple visual paradigm can be used to study the impact of tauopathy on intrinsic plasticity in awake animals, both at short (seconds and minutes) and longer (days) time scales. We have also shown that plasticity over longer time scales is markedly reduced in older animals, consistent with age-related reductions in ocular dominance plasticity in mice that have been documented previously[38,39]. In humans, repetitive visual stimulation produces a lasting enhancement of VEPs as measured by EEG[40,41], similar to SRP. This potentiation has been shown to be impaired in disorders that are thought to be associated with hypofunction of the NMDA receptor, like schizophrenia[42]. AD patients also show a deficit in NMDAR-dependent forms of cortical plasticity[43]. Assessing whether VEP potentiation is impaired in individuals with or at risk of AD, or in normal ageing, could be a subject for future studies. If true, it could provide a framework to assess disease progression in these patients that could be less invasive and cheaper than other biomarkers. Similarly, short-term visual adaptation effects are easy to measure in humans, both perceptually and via non-invasive EEG or fMRI measurements[44].

Visual deficits are common in dementia, and are particularly acute in patients with posterior cortical atrophy (PCA), which affects many sufferers of Alzheimer's disease (AD)[45]. The neural bases of these effects in PCA patients are thought to be tau-related pathological changes in posterior cortices such as the occipital lobe[46–48]. Recent large-scale characterisations of tau-patterning in AD show that a posterior, occipitotemporal subtype is as frequent as other subtypes[49]. Functional measurements of visual cortical plasticity may therefore be particularly relevant to these individuals.

Previous work in rTg4510 mice has shown a link between neurodegeneration and plasticity in high-level cortical circuits. rTg4510 mice show impaired long-term depression in perirhinal cortex synapses, which may underlie defects in long-term recognition memory[50]. Impairments in LTP are also observed in the hippocampus[51–53] and frontal cortex[54] of rTg4510 (and APP/PS1) mice, including changes in intrinsic membrane properties, depolarisation of the resting potential, increased excitability and changes in spiking dynamics. These cellular changes are accompanied by disturbed oscillations in the LFP and disordering of 'place cells' in hippocampus[55–57]. These deficits in both short- and medium-term neuronal plasticity are not simply a result of accumulation of insoluble NFT, and arise before pronounced changes in cellular morphology[54,58,59]. Our findings complement and extend these studies by showing that intrinsic LTP-like visual plasticity is also disrupted in rTg4510 animals in vivo, even at early stages of tauopathy. We have further shown that visual plasticity decreases with age even in Tau− and WT animals. Our results suggest that age should be taken into account when studying the impact of tauopathy on LTP-like effects in older animals. Finally, we have shown that there are substantial changes in plasticity in primary visual cortex, an area that contributes to functional responses in higher-order brain areas like the hippocampus, perirhinal, and prefrontal cortices. The contribution of primary sensory cortices may therefore need to be considered when studying the impact of tauopathy on plasticity in higher-order areas in vivo.

The functional effects observed at early stages of tauopathy in rTg4510 mice may be explained by synaptic dysfunction such as down-expression of NMDA and AMPA receptors[51,60,61], increased synaptic instability as a result of synaptic density reduction and increased dendritic spine turnover[62] or may reflect mislocalization of soluble tau to dendrites[63]. NMDAR and AMPAR play an important role in both synaptic plasticity and the expression of SRP, and loss of functional subunits might explain the reduction in SRP that we see in Tau+ mice. Synaptic instability is also likely to impair long-term plasticity processes. Our findings of impaired short-term adaptation effects in 5-month old Tau+ mice may reflect postsynaptic abnormalities as a result of tau accumulation in dendritic spines, or a reduction in the availability of excitatory synapses, which is then exacerbated in older animals. Future studies could utilise our findings in the visual cortex to study the relationship between plasticity impairments and synaptic dysfunction in neurodegeneration in vivo. The changes in short-term adaptation effects that we see in Tau+ mice provide a particularly attractive target for experimental measurements.

We note that retinal and optic nerve atrophy has been reported in late-stages (ca. 7.5 months) of rTg4510 tauopathy[64]. These changes in retinal structure may be expected to reduce the visual cortical response as measured by the VEP. However, we found that VEP amplitude was robust, even in 8-months old Tau+ animals. Tau+ mice also have altered circadian rhythms, with increased periods of wakefulness, less time in non-rapid-eye-movement sleep[65], and altered sharp-wave ripple dynamics[57]. These changes may contribute to the changes in long-term visual plasticity that we observed as sleep is important for SRP in mouse V1[18,28,66,67].

Recent work[68] shows that in the rTg4510 mouse model, a fragment of the *Fgf14* gene was replaced by the insertion of the P301L transgene (tau-Tg). This genetic disruption may influence the progress of neuronal loss and behavioural abnormalities alone or in combination with the expression of the mutant tau. Our experiments mitigate these potential offsite effects by comparing Tau− and Tau+ mice, that differ in mutant tau expression but not phenotype. In addition, our comparisons of Tau− with WT littermates reveal similar long-term plasticity and visual evoked behaviour. Interestingly, older (8 m) Tau− mice showed reduced adaptation, or short-term plasticity but WT animals did not. Adaptation effects are generally thought to be preserved with age in humans[69,70]. It is possible that reduced adaptation in Tau− animals could be an effect of the *Fgf14* deletion or it could be an effect of tau accumulation before the onset of the doxycycline treatment (which started at 2 months). If the latter is true, this would render adaptation as a more sensitive assay to assess functional changes in neurodegeneration in older animals.

Previous studies have reported alterations in rTg4510 mouse behaviour: Tau+, and to a lesser extent, Tau− mice tend to show hyperactivity and motor stereotypy, whilst maintaining good motor coordination[13,22,23,65,71]. These mice do not appear to respond to novelty, and have impaired spatial working memory[13,22]. In our experiments on head-fixed animals, Tau+ animals reacted less to the appearance of a new visual stimulus than did their Tau− or WT counterparts, consistent with reduced novelty responses. Reduced behavioural responses to novelty in Tau+ mice may indicate a deficit in other regions of the visual pathway, such as the Superior Colliculus (SC). However, histological analysis shows little accumulation of phosphorylated tau in the SC of rTg4510 animals[64]. The insensitivity to novelty may partly reflect a lack of arousal or a deficit in attention, which may, in turn, indicate deficiencies in noradrenaline circuits that appear critical for these processes[72]. The CamKIIa promoter that drives transgene expression in the rTg4510 mouse is likely to be

expressed in locus coeruleus[73], and it is, therefore, possible that the noradrenalin input to cortex is disrupted in Tau+ mice.

Plasticity is a hallmark of neuronal function, important for learning and memory. Neurodegenerative diseases like AD not only disrupt neuronal structure and function, but erode the flexibility of neurons and circuits. Synaptic dysfunction occurs early in the course of AD and may contribute to the failure of synaptic plasticity and the development of the disease. We verified the effect of tauopathy on an intrinsic form of sensory learning and memory in V1 of mouse. We found impaired visual plasticity even at early stages of tauopathy, before a substantial neuronal loss occurs. Our measurements offer a simple and direct readout of intrinsic plasticity in degenerating brain circuits, and a potential target for understanding, detecting, and tracking that neurodegeneration, in humans, as well as in animal models.

## Methods

**Animal experiments**. All experiments were performed in accordance with the Animals (Scientific Procedures) Act 1986 (United Kingdom) and Home Office (United Kingdom) approved project and personal licenses. The experiments were approved by the University College London Animal Welfare Ethical Review Board under Project License 70/8637.

The generation of rTg4510 transgenic mice was performed as described previously[11,13]. A total of 50 male transgenic mice and 16 wild-type (WT) littermates were obtained at approximately 7 weeks of age from Eli Lilly and Company (Windlesham, UK) via Envigo (Loughborough, UK). At 8 weeks of age, to suppress tau expression, 25 of the 50 transgenic mice were treated with Doxycycline, which included four 10 mg/kg bolus oral doses of doxycycline (Sigma) in 5% glucose solution by oral gavage across 4 days, followed by *ad libitum* access to Teklad base diet containing 200 ppm doxycycline (Envigo) for the duration of the experiment. The mice in this group were designated 'Tau−' animals. The remaining 25 animals, designated as 'Tau+', and the WT animals received 4 oral doses of the vehicle (5% glucose) and had *ad libitum* access to standard feed for the duration of the experiment. All animals also had *ad libitum* access to water. Mice were subsequently divided into two cohorts. One cohort was tested at approximately 5 months (22–26 weeks, 12 Tau−, 12 Tau+, 6 WT), and another cohort was tested at approximately 8 months of age (31–35 weeks, 13 Tau−, 13 Tau+, 10 WT). Mice were group housed to a maximum of 5 individuals per cage until 3 days before surgery, when they were separated into individual cages. All animals were kept under a 12 h light/dark cycle, and both behavioural and electrophysiological recordings were carried out during the dark phase of the cycle.

**Surgery**. Mice were anaesthetised for surgery with 3% isoflurane in $O_2$. Preoperative analgesia (Carprieve, 5 mg/kg) was given subcutaneously and lubricant ophthalmic ointment was applied. Anaesthesia was maintained with 1–2% isoflurane in $O_2$ and the depth was monitored by the absence of pinch-withdrawal reflex and breathing rate. The body temperature was maintained using a heating blanket. A small craniotomy hole (<1 mm²) was made over the right primary visual cortex (2.8 mm lateral and 0.5 mm anterior from lambda) and a chronic LFP recording electrode (Bear lab chronic microelectrode Monopolar 30070, FHC, USA) was implanted approximately 450 μm below the cortical surface. Stereotaxic coordinates were adjusted for 8 m old Tau+ mice to 2.7 mm lateral and 0.55 mm anterior from lambda and 400 μm depth to account for the reduction in brain size in animals with advanced tauopathy. A ground screw was implanted over the left prefrontal cortex and a custom-built stainless-steel metal plate was affixed on the skull. Dental cement (Super-Bond C&B, Sun Medical) was used to cover the skull, ground screw, and metal plate, enclosing and stabilising the electrode. Analgesic treatment (Metacam, Boehringer Ingelheim, 1 mg/kg) mixed in condensed milk was provided orally for three days after the surgery. Mice recovered for at least 7 days before the first recording session.

**Visual stimulus presentation and experimental setup**. Visual stimuli consisted of full-field, 100% contrast sinusoidal gratings generated using BonVision[74], presented on a γ-corrected computer monitor (Iiyama ProLite EE1890SD). The gratings were presented in a circular aperture with hard edges, outside of which the monitor was held at the mean luminance ('grey screen'). The grating was oriented at either −45° or 45° from vertical, and reversed contrast (flickered) at a frequency of 1.95 Hz. The display was placed 15 cm from—and normal to—the mouse and centred on the left visual field. The stimulus was warped to maintain visual angle across the monitor.

One week after the surgery, mice were placed on a styrofoam wheel with a grey screen and habituated over 5 days to the experimental set-up by progressively increasing the time spent head-fixed, from ~5 to 30 min. Mice were allowed to run on the wheel, and their speed was recorded using a rotary encoder. Pupils were imaged using an infrared camera (DMK 22BUC03, ImagingSource; 30 Hz) focused on the left eye through a zoom lens (Computar MLH-10X Macro Zoom Lens), and

acquired by the same computer that presented the visual stimulus. Pupil estimates (position, diameter) were tracked online using custom routines in Bonsai. At the beginning of each recording session, mice were presented with a grey screen for 3–5 min. On the first day of recordings, mice were presented with five blocks of a grating oriented at 45° and five blocks of a grating oriented at −45°, alternating between the two. Each block consisted of 200 continuous reversals, and were separated by 30 s, during which the monitor was held at the mean luminance. For six animals (2 WT, 2 Tau− and 2 Tau+), each block consisted of 400 phase reversals. On days 2–8, mice were presented with 10 blocks of a single oriented grating (familiar stimulus), that was either 45° or −45°, randomly assigned for each animal (counterbalanced between groups). The last day of recordings, day 9, was similar to day 1. For days 1 and 9, whether the first block presented would be the familiar or the unfamiliar stimulus (presented only on the first and last day of recordings), was randomly assigned for each animal.

**Neural recordings**. Signals from the recording electrode were acquired, digitised, and filtered using an OpenEphys acquisition board connected to a different computer from that used to generate the visual stimulus. The electrophysiological and rotary wheel signals were sampled at 30 kHz. These data were synchronised with pupil video recordings and visual stimulus via the signal of a photodiode (PDA25K2, Thorlabs, Inc., USA) that monitored timing pulses on a small corner of the monitor shielded from the animal.

**Data analysis**. All data were analysed using custom software written in MATLAB (MathWorks). Neural and wheel data were filtered with an 8th order Chebyshev Type I lowpass filter and downsampled to 1 kHz.

*VEP analysis*. VEPs were averaged across all phase reversals and blocks for each stimulus on each day. To estimate the VEP amplitude, the LFP signal was filtered using a second order bandpass filter with a 0.3 Hz low cut and 50 Hz high cut frequency. The negative trough was defined as the minimum value within the first 150 ms after stimulus reversal, and the positive peak was defined as the maximum value within the first 250 ms. The amplitude was defined as the difference between this trough and peak.

*Pupil data*. Eye blinks were removed by identifying any points that were two times above the variance of the eye position. Removed values were replaced using nearest neighbour interpolation. Pupil area (in pixels) was converted to $mm^2$ by multiplying with the square of the camera resolution (in mm/pixel). Responses were then normalised to the average pupil area in the 2 min before the stimulus onset, where animals were viewing a grey screen.

*Wheel data*. We estimated the speed and direction of the rotating wheel using a quadrature encoder. Rotations were converted to speed by multiplying with the wheel circumference and dividing by the encoder resolution. Speed was then smoothed using a gaussian filter with a 50 ms window.

*Regression*. We fitted an elastic net regularisation model to predict the VEP amplitude (target variable) for each animal using days, block number within each day, movement speed, and pupil area as regressors (also known as features or input variables). To do so, we used MATLAB's function *lasso*. The model assigns weights to the defined features (regressors) in order to predict the output (VEP amplitude). Regressors and predicted (target) value were normalised to range between 0 and 1 before fitting. Elastic net regularisation encourages sparse solutions, i.e., solutions where some of the learned variable weight coefficients are equal to 0, improving model generalisation, and hence prediction accuracy. Model sparsity is determined by elastic net's two regularisation hyperparameters. Because we don't know the optimal values of these hyperparameters, we explore the entire regularisation path. Here we used $L \leq 10,000$ values of the regularisation hyperparameter, fixing the ratio of the L1 over L2-norm regularisers to 2. We estimated the impact $M$ of each regressor $r$, based on the derived regression weights, as the percentage of contribution to the estimates of the VEP amplitude across the entire regularisation path. This was defined as:

$$M(r_i) = \sum_{l=1}^{L}\sum_{t=1}^{T} w_{i,l} r_{i,t} \bigg/ \sum_{l=1}^{L}\sum_{t=1}^{T} \hat{y}_{l,t} \qquad (1)$$

Where $L$ are the values used for the regularisation hyperparameter, $t$ denotes each time there is a stimulus reversal ($t = 1 : number\ of\ reversal\ each\ day * number\ of\ days$), $w$ is the learned weight, $r_i$ denotes each respective regressor (where $i$ is either days, block number, speed, pupil area) and $y$ is the predicted value of the elastic net model. Note that if two or more regressors are correlated, the regularisation model can assign weights to one or both regressors.

*Visual adaptation*. To estimate the adaptation effect for each animal, for each day, the VEP amplitude was calculated within a block using a step of 20 reversals. The mean amplitude of the last 200 reversals was subtracted from each block and amplitudes were averaged over blocks 2–10, or blocks 2–5, or blocks 6–10. A decaying exponential function was fitted to the averaged data using least squares. The exponential time constant for each animal was fixed to $\tau = 8.4$ reversals. This

value was calculated by fitting the exponential to the mean trace obtained by averaging over all animals, for days 2–8, and blocks 2–10.

**Brain samples**. Mice were euthanised by overdose injection of pentobarbital (intraperitoneal) and perfused with phosphate-buffered saline (PBS). Brains were removed, weighed, and the right hemisphere was fixed in 10% buffered formalin until processed (7–13 months) for immunohistochemistry pathology assessment.

**Histopathology**. Immunohistochemistry was performed for all mice. The brains were immersed in PBS that contained 30% sucrose, and subsequently cut into 40 μm parasagittal sections on a cryostat (Leica CM1520). Antigen retrieval was achieved through heating sections in citrate buffer (pH 6.0; Vector Labs) in an oven at 60 °C overnight. Slides were treated with 0.3% hydrogen peroxide solution (3% in distilled water) for 10 min to eliminate endogenous peroxidase activity, and subsequently washed with PBS with 0.5% Triton X-100. Immunohistochemistry was performed using a primary antibody for tau phosphorylated at serine 202 (mouse monoclonal AT8, 1:1000, Thermo Fisher Scientific). The Mouse on Mouse (MOM) Detection Kit (Vector Labs, BMK-2202) was used for staining, with buffers prepared as described in standard protocol supplied with the kit. After rinsing, slides were treated for 5 min with the chromogen 3,3′-diaminobenzidine (DAB; Vector Laboratories, SK-4105) to allow visualisation. The slides were then coverslipped with Shandon ClearVue Mountant XYL (Thermofisher) and digitised using a Leica Microscope (DMi8 S) coupled with a Leica camera (DFC7000 GT; Leica Micro-systems). The Fiji image processing package was used to view the digitised tissue sections[75]. At least 3 regions of interest (ROIs) of the same size were selected from approximately the same brain location in V1 and hippocampus for each animal. In cases where V1 sections were fragmented ($n = 12$), additional ROIs were selected from elsewhere in the cortex. To assess the tau burden, a normal distribution was fit to the image of each ROI, and the mean and standard deviation were obtained and averaged over all ROIs for each animal (Supplementary Fig. 1). To compare the tau burden in V1 and hippocampus, the mean values obtained from the fitted distributions were zscored and sign inverted (Fig. 1). These analyses were performed in a blinded fashion. The location and depth of the recording electrode were confirmed by imaging the electrode track in histological sections. The electrophysiological data from one 5 m old Tau+ mouse were removed from the analysis because histological analysis revealed that the electrode was implanted too deep.

**Statistics and reproducibility**. All data are presented as a mean ± standard error of the mean (SEM). Statistical comparisons were performed in MATLAB and SPSS (IBM). A Student's t-test, two-way, or repeated-measures ANOVA was applied for comparisons as described in the main text. $p < 0.05$ was used as the significance threshold. Exact $p$ values are given.

**Reporting summary**. Further information on research design is available in the Nature Research Reporting Summary linked to this article.

## Data availability

The datasets generated during the current study are available from the corresponding author on reasonable request. The source data underlying Figs. 1–6 are available in Supplementary Data 1–6.

## Code availability

The code generated during the current study is available from the corresponding author on reasonable request.

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

## Acknowledgements

We thank Francesca Cacucci (F.C.) for comments on the manuscript. We thank Zeshan Ahmed, and Anthony Blockeel for useful advice and discussions. We thank Henrik Singmann for useful advice on the statistical analyses. This work was supported by the Medical Research Council grant (R023808) to S.G.S., A.B.S. and F.C., a Biotechnology and Biological Sciences Research Council grant (R004765) to S.G.S. and A.B.S., a Sir Henry Dale Fellowship from the Wellcome Trust and Royal Society (200501) to A.B.S, and an International Collaboration Award to S.G.S. (with Adam Kohn) from the Stavros Niarchos Foundation/Research to Prevent Blindness.

## Author contributions

Conceptualisation, A.P., A.B.S., and S.G.S.; methodology, A.P., F.R.R., A.B.S., and S.G.S.; investigation, A.P., F.R.R., and J.H.; validation, formal analysis, data curation, A.P., F.R.R., S.G.S.; writing—original draft, A.P. and F.R.R.; writing—review & editing, A.P., F.R.R., A.B.S., and S.G.S.; visualisation, A.P., F.R.R., A.B.S., and S.G.S.; funding acquisition, A.B.S., and S.G.S.; resources, A.B.S., K.G.P., and S.G.S.; supervision, A.B.S. and S.G.S.

## Competing interests

The authors declare no competing interests.
