## [Transparent Peer Review File · Communications Biology]

Reviewers' comments:

Reviewer #1 (Remarks to the Author):

This manuscript by Papanikolaou et al. aims to close a gap. While numerous in vitro and in vivo studies have shown that a progressing tau pathology leads to an impairment of artificially induced synaptic plasticity in the hippocampus and cortical regions of AD rodent models, this work presents for the first time evidence that this also applies to intrinsically triggered synaptic plasticity. The authors used the rTg4510 transgenic mouse model to investigate whether tau-driven neurodegeneration impairs a simple form of intrinsic plasticity in the visual system at both long (days) and short (minutes) timescales. In the ms, two stages of pathology are compared: (i) 5 months of age, at which already tangle-like PH1-positive inclusions, Bielschowsky silver-positive NFTs, reduced activity of the neocortical network, impaired LTP in vitro and deficits in cognitive tests have been reported, as referred to by the authors and also summarized in ALZFORUM. (ii) 8 months of age with further aggravated tau-pathology and beginning cortical cell loss (Spires et al. 2006). When the animals were repeatedly exposed to a stimulus over 9 days and local field potentials were recorded in the primary visual cortex, the authors report that both short- and long-term visual plasticity were already deteriorated at early stages of neurodegeneration as compared to animals in which the transgene expression was suppressed. In older animals, this functional deficit was even larger. In line with the electrophysiological findings, visually evoked behaviours were found to be disrupted in both younger and older mice expressing the P301L transgene.

The main findings are interesting and the ms is well written, but the description of the methods and experiments lacks detail in some cases. My main concern is that the analysis showed that the VEP amplitude is dependent on the behavioural state. This applies not only to the basal state but also to plasticity measures (as shown in Fig. S5) which are usually more sensitive to behavioural modulation than basic electrophysiological readouts. It is relevant in this context, that abnormal locomotion behaviours have been reported for Tau+ rTg4510 mice as cited in the ms. Thus, a different frequency and duration of such behavioural epochs may have a decisive influence on the outcome of experiments that primarily focussed on intrinsic plasticity in the visual system. In this case, found effects would be severely modulated or even caused by Tau pathology in regions outside the visual system that control these modulating behaviours. Hence, I recommend to re-analyse further data sets (e.g. long-term plasticity data) for which this could be relevant to dissect which effects can be ascribed to Tau pathology in the visual system and which could be rather indirect effects of Tau pathology in regions that control behaviours such a locomotion. For example, I recommend to check the data shown in Figure S5 for differences between Tau- and Tau+ mice.

Below are a number of additional concerns, questions and suggestions:

Introduction

line 7 The authors state: "In vivo measurements in these mouse models also show disruption of artificially-induced LTP^{5,6}". However, these experiments were carried out on urethane anaesthetized adult male Wistar rats, not on mice as stated in these papers. Thus, rephrase or refer to other papers.

Results

Given that it is not outlined in Methods according to which criteria the electrode positioning was optimized/ adjusted, I would like to know how the different size of the brains between the examined age groups and the reduction in brain size of tau+ and Tau- mice compared to WT, which affects the values of the optimal stereotaxic coordinates, was taken into consideration?

line 55ff: The authors used the amplitude of the VEP as the only measure that was analysed. Field potentials are the integral result of the superposition of various components from different sources.

Therefore, it would be more convincing if the conclusions are not only based on just one measure. Why were no other measures used in addition, such as the decay slope following the peak or the latency to peak? For example, in the average traces of 8-month-old animals depicted in Fig. 1E, the latency seems to be about 10% greater in Tau+ animals. Furthermore, the authors mention on line 63 that the VEP signal was more sustained in 8-month-old animals. I guess this statement refers to the slower decay of the potential after the peak. It might be worthwhile to analyse this obvious difference quantitatively by using the decay slope or the decay time constant after the peak.

line 56ff: The amplitudes of the VEPs were obtained by averaging across all phase reversals and blocks for each stimulus on each day as described in methods. In so doing, the authors did not find significant differences in the average amplitudes between genotypes or age groups. However, later they describe that the behavioural state (running vs. stationary behaviour) had a significant effect on VEPs. As mentioned already above, the effects of Tau pathology on basal LFPs would probably become better discernible when the effects of behavioural modulation would be considered in the analysis. Did you perform the analysis described in line 55ff also with the VEP data of Tau- and Tau+ mice separately grouped according whether they were collected during running epochs or stationary behaviour? Did you also carry out the analysis described in line 55ff with the VEP data from Tau- and Tau + mice grouped separately according to running epochs or stationary behaviour? Did you find typical differences in the shape of VEPs when potentials of Tau- and Tau+ mice collected during running epochs were compared with stationary phases?

line 103ff: The reduced and slower Stimulus-Response Potentiation in Tau+ as compared to Tau- mice is an interesting finding. However, the difference in SRP of Tau+ and Tau- mice depends very much on the way the data are analysed. It is not explained in the methods what the shaded area on both sides of the VEP time course in figures 2 and S2 stands for, but it is most likely a sort of confidence band (95% ?; please explain what is shown). Furthermore, the two graphs for the VEP amplitude in Fig.2B and Fig.S2a display apparently the same data. However, while the confidence band in Fig.2B with 'normalized' amplitude is of lean appearance indicating no overlap and a possible statistical difference between the amplitudes of both genotypes between days 2 and 4, the confidence band in Fig.S2a (displaying the absolute values of VEP amplitudes) is completely and broadly overlapping from day 1 to day 9. Even if the starting points of the curves of the two genotypes in Fig.S2a would be shifted to be completely on top of each other on day 1, a strong overlap of the confidence bands would remain. (To make clear what I mean: When I recalculate/estimate the values of the amplitude curves for the 5-month-old animals from Figures 2B and S2a, I get for example a difference of the genotype means of about 45 μV for day 4 in the normalized values in Fig.2B but only of about 23 μV for the same day in the absolute values in Fig.S2a. Thus, what is wrong in the presentation of these data and why are the normalized amplitudes shown as time course in Fig 2B on the left, but the absolute amplitudes used for the statistics displayed in Fig.2 C-E?

Given the similarity of the curves of the absolute VEP amplitudes shown in Fig.S2a and the broad overlap of their confidence band, the statement that visual plasticity is disrupted at this early stage (i.e. 5 months of age) of neurodegeneration should be toned down.

line 134: Again, the normalized values are shown in subfigure B on the left, but the absolute values are used for statistics and shown in C and D. This is not consistent and misleading in the view of the reviewer!

Figure 4: The scheme in Fig.4A seems to indicate that the individual example for a short-term response of the VEP amplitude from a 5-month-old mouse shown in B below is taken from day 3. If this supposition is not correct, please explain what the dotted lines mean. Furthermore, indicate clearly whether the example for an 8-month-old mouse is also from day 3.

line 182: The statistical test used to compare the cumulative histograms of the fitted amplitudes should be given in the Methods and here in the text.

line 219: You refer here to Fig.4 for within day variations, is this correct?

line 230: Based on the data of your elastic net regularization model, you conclude here that behaviour cannot explain the VEP increase across days, supporting the presence of plasticity. For most of the readers this model will be a black box and difficult to assess. Importantly, the analysis showed that the VEP amplitude is dependent on the behavioural state. This applies not only to the basal state but also to plasticity measures (as shown in Fig. S5) because they are usually more sensitive to behavioural modulation than basic electrophysiological readouts. As mentioned already before, I recommend to re-analyse further sets of long-term plasticity data, e.g. the data shown in Figure S5 for differences between Tau- and Tau+ mice.

Discussion

line 303: "There is some evidence..." Please explain here in more detail how the impairment of inhibitory function early in neurodegeneration may result in reduced SRP in Tau+ animals.

line 327: As mentioned above, the proven direct effects of Tau pathology on long-term plasticity are rather mild in the current version of the ms. Therefore, phrases like "allows easy characterization of the impact of neurodegeneration on intrinsic plasticity" should be toned down, as long no additional evidence and facts are provided.

line 333: In this context, published data about NMDAR function in rTg4510 mice should be mentioned and discussed [e.g. Ma et al. (2020), *Curr Med Sci* 40:1031-1039].

line 363ff: It is not clear to the reviewer, why increased synaptic instability is chosen here as the prime mechanism to explain the functional deficits observed by the authors in rTg4510 mice at early stages of neurodegeneration.

Methods

line 422: From the present description of the experiments it is difficult to deduce in detail when which experiments were carried out with which animals and whether certain topics were investigated with sub-data sets from the same experiments. Therefore, please provide a clear diagram of the timeline of the individual experiments.

Reviewer #2 (Remarks to the Author):

In this manuscript, Papanikolaou et al. examine the impact of Tauopathy on visual cortical function and plasticity, using the TG4510 mouse model. In freely behaving mice, they assess visual cortical response to a simple, grating stimulus and measure visual cortical plasticity across different timescales as mice become familiar with a specific orientation of stimulus. They measure visual cortical function and plasticity at two different ages, comparing littermate controls with mutant Tau over-expressors at 5 months of age and 8 months of age. They demonstrate that, while basal visual cortical responses seem largely unaffected when measured with electrophysiology, there is nevertheless a compromised behavioural response to a visual stimulus, even in young Tau+ mice. They then reveal that both short-term visual cortical plasticity and long-term plasticity over days is compromised by mutant Tau+ over-expression. Interestingly, this plasticity is generally age-

dependent, as it is drastically reduced by 8 months in both Tau- and Tau+ mice. Nevertheless, residual plasticity is also reduced in the Tau+ mice. In general, I think these findings are important because they demonstrate a means to measure plasticity as an animal learns in disease models. This represents substantial progress over the more standard approach of experimental tests of synaptic plasticity such as LTP and LTD, which may be conducted in vivo or ex vivo but, in both cases, are induced by extremely artificial means. An additional future benefit of this work may be that some similar methods could be applied in human subjects. I do have some concerns about the interpretation of the findings in this manuscript that I will now discuss:

- 1) The study is designed to assess the impact of neurodegeneration on plasticity, and this is the way that both the abstract and introduction are pitched. However, the ages of the mice that have been selected here are both prior to the documented neurodegeneration threshold of 8.5 months, as stated by the authors in their own introduction. Why not, therefore, include a group that is considerably past this point and with clear neurodegeneration.
- 2) It would be helpful to have some indication of the degree of neurodegeneration/tauopathy in visual cortex of the mice used here. If the authors still have the brains of the mice used, then that could be evaluated. There might be an opportunity to compare the degree of neurodegeneration or tauopathy with the impact on cortical plasticity on a mouse-by-mouse basis. Alternatively, the emphasis of the manuscript could be switched to tauopathy rather than neurodegeneration.
- 3) The mismatch between the intact visual cortical response and the lost visual-induced behavioural response may indicate that the behaviors tested here require normal function in other visual pathways/regions, perhaps including the superior colliculus. I suggest it would be worth discussing this possibility or providing a strong justification for the behaviors that have been used as assays of visual cortical function.
- 4) A growing view among those that work on dementia is that synaptic dysfunction precedes neurodegeneration by quite some time and that it may serve as the origin for mild cognitive impairments or even more prodromal symptoms of dementia. Therefore, to me, the most exciting aspect of the findings in this manuscript is that cortical plasticity deficits manifest in tauopathy could be measured in the living animal (and perhaps eventually human) at a very early age. That may present a biomarker with predictive value for Dementia. This concept could be given some emphasis in the manuscript.
- 5) It seems likely that minimal neurodegeneration has occurred by the time these mice are showing this pronounced deficit in plasticity at 5 months. It would be very interesting to see if the deficit is present at a much earlier age of, say, 1 or 2 months. Perhaps the overexpression of mutant tau is causing the synaptic plasticity deficit rather than any degenerative process.
- 6) The authors raise the issue of inhibition as a key contributor to stimulus-selective response plasticity and that this may be aberrant in the Tauopathy model. However, as the authors themselves point out, the mutant Tau should be over-expressed under the CaMKII promoter, which should not produce expression in cortical inhibitory neurons. Moreover, a loss of cortical inhibition would surely affect visual cortical response magnitude. In fact, I would expect to see greatly increased VEPs. This issue could do with consideration in the discussion section.

02 November 2021

We would like to thank the reviewers for highlighting the contribution of our manuscript and providing insightful comments and suggestions to improve it. In light of the comments, we have re-analysed all data as the reviewers suggested and have expanded the methods and discussion sections. We greatly appreciate the constructive feedback and feel that the manuscript has greatly improved based on the suggestions.

Our answers to specific points are detailed below.

Sincerely,
Amalia Papanikolaou, Fabio Rodrigues, Aman Saleem, Samuel Solomon

Reviewer #1:

This manuscript by Papanikolaou et al. aims to close a gap. While numerous in vitro and in vivo studies have shown that a progressing tau pathology leads to an impairment of artificially induced synaptic plasticity in the hippocampus and cortical regions of AD rodent models, this work presents for the first time evidence that this also applies to intrinsically triggered synaptic plasticity. The authors used the rTg4510 transgenic mouse model to investigate whether tau-driven neurodegeneration impairs a simple form of intrinsic plasticity in the visual system at both long (days) and short (minutes) timescales. In the ms, two stages of pathology are compared: (i) 5 months of age, at which already tangle-like PH1-positive inclusions, Bielschowsky silver-positive NFTs, reduced activity of the neocortical network, impaired LTP in vitro and deficits in cognitive tests have been reported, as referred to by the authors and also summarized in ALZFORUM. (ii) 8 months of age with further aggravated tau-pathology and beginning cortical cell loss (Spires et al. 2006). When the animals were repeatedly exposed to a stimulus over 9 days and local field potentials were recorded in the primary visual cortex, the authors report that both short- and long-term visual plasticity were already deteriorated at early stages of neurodegeneration as compared to animals in which the transgene expression was suppressed. In older animals, this functional deficit was even larger. In line with the electrophysiological findings, visually evoked behaviours were found to be disrupted in both younger and older mice expressing the P301L transgene.

We thank the reviewer for their interest and for pointing out the contribution of our paper.

(R1.1) The main findings are interesting and the ms is well written, but the description of the methods and experiments lacks detail in some cases. My main concern is that the analysis showed that the VEP amplitude is dependent on the behavioural state. This applies not only to the basal state but also to plasticity measures (as shown in Fig. S5) which are usually more sensitive to behavioural modulation than basic electrophysiological readouts. It is relevant in this context, that abnormal locomotion behaviours have been reported for Tau+ rTg4510 mice as cited in the ms. Thus, a different frequency and duration of such behavioural epochs may have a decisive influence

on the outcome of experiments that primarily focussed on intrinsic plasticity in the visual system. In this case, found effects would be severely modulated or even caused by Tau pathology in regions outside the visual system that control these modulating behaviours. Hence, I recommend to re-analyse further data sets (e.g. long-term plasticity data) for which this could be relevant to dissect which effects can be ascribed to Tau pathology in the visual system and which could be rather indirect effects of Tau pathology in regions that control behaviours such as locomotion. For example, I recommend to check the data shown in Figure S5 for differences between Tau- and Tau+ mice.

We reanalysed the data in two different ways in order to understand whether differences in plasticity between Tau- and Tau+ animals can be explained by differences in locomotor behaviour (lines 244-274). First, we asked if running had an acute effect on VEP responses and, related to this, if observed SRP effects were present when we only considered the stationary condition. Second, we asked if the amount of running on one day influenced SRP effects on subsequent days.

We found that the observed SRP effects are present and even larger when we considered only the stationary sessions. We refined our analyses by only considering VEP responses during either stationary (speed < 1cm/s) or running (speed > 1cm/s) epochs and we compared the change in VEP amplitude from day 1 for Tau+ and Tau- mice. The differences in SRP between 5 month old Tau- and Tau+ mice were more pronounced when we considered only stationary epochs (Fig. 5C, S5A-C). Repeated measures ANOVA showed a significant day by genotype interaction ($F=2.07$, $p=0.041$) for the stationary VEP amplitude; we also compared the change in VEP amplitude (Fig. 5C) between Tau- and Tau+ mice on each day for stationary epochs and found a significant difference for days 2,3,4,7,8 (day 2: $p=0.02$, day 3: $p=0.002$, day 4: $p=0.022$, day 7: $p=0.015$, day 8: $p=0.019$, Tukey's).

We next estimated if the amount of time spent running in a session had an effect on the potentiation observed on the following day. We first calculated the amount of time animals spent running during stimulus presentation. We found that Tau- animals were more active during stimulus presentation compared to Tau+ mice, at both 5m (Tau-: $17\pm 16.8\%$, Tau+: $10\pm 8.7\%$; $p=2.7\cdot 10^{-4}$, t-test) and 8m (Tau-: $33\pm 31\%$, Tau+: $13\pm 10\%$; $p=3.2\cdot 10^{-10}$). We therefore asked whether the amount of running correlated with increased plasticity. We found no correlation between the amount of time animals spend running during stimulus presentation on one day and the amount of SRP on the next day at 5 months ($r=0.07$, $p=0.29$). We also found no correlation between the average running activity of animals and the total amount of SRP found by day 9 ($r=-0.26$, $p=0.16$). Therefore, running behaviour could not predict visual plasticity in 5 month old animals. At 8 months, there was a small positive correlation between the amount of time animals spend running during stimulus presentation and the amount of SRP on the next day ($r=0.16$, $p=0.006$), but visual plasticity was much less in both Tau- and Tau+ animals at 8 months.

We would like to note that while re-analysing the data, we discovered a multiplication factor in our estimates of running speed that occurred in the conversion of angular wheel rotation to cm/s. This has not affected our main conclusions, because we had chosen a speed threshold based on the observed speed distributions in these animals. We have now corrected this in all of our analyses.

Below are a number of additional concerns, questions and suggestions:

Introduction

(R1.2) line 7 The authors state: "In vivo measurements in these mouse models also show disruption of artificially-induced LTP5,6". However, these experiments were carried out on urethane anaesthetized adult male Wistar rats, not on mice as stated in these papers. Thus, rephrase or refer to other papers.

Thank you for picking up this error. We now make sure the citations are to the relevant papers of mouse models we meant to include.

Results

(R1.3) Given that it is not outlined in Methods according to which criteria the electrode positioning was optimized/ adjusted, I would like to know how the different size of the brains between the examined age groups and the reduction in brain size of tau+ and Tau- mice compared to WT, which affects the values of the optimal stereotaxic coordinates, was taken into consideration?

We thank the reviewer for pointing this out. We had indeed accounted for the reduction in brain size in animals with advanced tauopathy by adjusting the stereotaxic coordinates during electrode implantation. We now include this information in the methods section (lines 498-500) and apologise for its earlier omission. Specifically, the coordinates used for 8m Tau+ mice were 2.7 mm lateral, 0.55 mm anterior from lambda, and 0.4 mm from the cortical surface – compared to 2.8 mm lateral, 0.5 mm anterior from lambda, and 0.45 mm from the cortical surface for all other animals. In the process of preparing this response we have reanalysed the histological slices around the electrode tracks and determined that in one Tau+ 5 month old animal the electrode was placed too deep in V1 (lines 597-600). We have now removed the electrophysiological data from that animal and reanalysed all data. Removal of this animal did not change our conclusions.

(R1.4) line 55ff: The authors used the amplitude of the VEP as the only measure that was analysed. Field potentials are the integral result of the superposition of various components from different sources. Therefore, it would be more convincing if the conclusions are not only based on just one measure. Why were no other measures used in addition, such as the decay slope following the peak or the latency to peak? For example, in the average traces of 8-month-old animals depicted in Fig. 1E, the latency seems to be about 10% greater in Tau+ animals. Furthermore, the authors mention on line 63 that the VEP signal was more sustained in 8-month-old animals. I guess this statement refers to the slower decay of the potential after the peak. It might be worthwhile to analyse this obvious difference quantitatively by using the decay slope or the decay time constant after the peak.

We concentrated on the VEP amplitude in the manuscript as it is the standard metric used in previous work on SRP. In supplementary data, we also analyse the positive and negative components that contribute to this (Fig. S2). As suggested by the reviewer, to provide a more complete characterisation of the VEP we now also provide: the time-to-peak since stimulus onset, the width-at-half-maximum for positive and negative peaks, and the decay slope following the positive peak. We analysed VEP waveforms on day 1 and found no significant differences between Tau- and Tau+, at 5m or 8m, for any measure (Fig. 1; lines 61-66). We do find a significant difference between 5m and 8m animals for the average VEP amplitude, the amplitude of the negative peak, and the time to negative and positive peaks. We summarise these findings in lines 61-66 and Table S1. We also calculated these measures for days 2-9 during the SRP paradigm (lines 133-136). We found a significant increase over days for the time to negative peak ($F=4.3$, $p=9 \times 10^{-5}$), time to positive peak ($F=6.1$, $p=6.7 \times 10^{-7}$) and the width at half maximum of the positive peak ($F=8.3$, $p=1.8 \times 10^{-9}$), but there was no significant difference between groups (day \times genotype: time to positive peak: $F=0.26$, $p=0.97$, time to negative peak: $F=0.85$, $p=0.56$, width of positive peak: $F=0.54$, $p=0.82$). Therefore, differences in SRP between Tau- and Tau+ animals are mainly characterised by changes in the amplitude of the VEP signal (particularly of the positive peak).

(R1.5) line 56ff: The amplitudes of the VEPs were obtained by averaging across all phase reversals and blocks for each stimulus on each day as described in methods. In so doing, the authors did not find significant differences in the average amplitudes between genotypes or age groups. However, later they describe that the behavioural state (running vs. stationary behaviour) had a significant effect on VEPs. As mentioned already above, the effects of Tau pathology on basal LFPs would probably become better discernible when the effects of behavioural modulation would be considered in the analysis. Did you perform the analysis described in line 55ff also with the VEP data of Tau- and Tau+ mice separately grouped according whether they were collected during running epochs or stationary behaviour? Did you also carry out the analysis described in line 55ff with the VEP data from Tau- and Tau + mice grouped separately according to running epochs or stationary behaviour?

Did you find typical differences in the shape of VEPs when potentials of Tau- and Tau+ mice collected during running epochs were compared with stationary phases?

We have now compared VEP parameters described in response to R1.4 above, between Tau+ and Tau- animals during stationary or running epochs (Table S1). We found that differences between 5m and 8m animals were evident during stationary epochs. The VEP signal during running epochs was reduced in all animals, and statistical differences were diminished.

(R1.6) line 103ff: The reduced and slower Stimulus-Response Potentiation in Tau+ as compared to Tau- mice is an interesting finding. However, the difference in SRP of Tau+ and Tau- mice depends very much on the way the data are analysed. It is not explained in the methods what the shaded area on both sides of the VEP time course in figures 2 and S2 stands for, but it is most likely a sort of confidence band (95% ?; please explain what is shown).

We apologise as this information was buried in the Methods section. The shaded error bars throughout the manuscript indicate the standard error of the mean. We have now clarified that in all relevant figure legends.

(R1.7) Furthermore, the two graphs for the VEP amplitude in Fig.2B and Fig.S2a display apparently the same data. However, while the confidence band in Fig.2B with 'normalized' amplitude is of lean appearance indicating no overlap and a possible statistical difference between the amplitudes of both genotypes between days 2 and 4, the confidence band in Fig.S2a (displaying the absolute values of VEP amplitudes) is completely and broadly overlapping from day 1 to day 9. Even if the starting points of the curves of the two genotypes in Fig.S2a would be shifted to be completely on top of each other on day 1, a strong overlap of the confidence bands would remain. (To make clear what I mean: When I recalculate/estimate the values of the amplitude curves for the 5-month-old animals from Figures 2B and S2a, I get for example a difference of the genotype means of about 45 μ V for day 4 in the normalized values in Fig.2B but only of about 23 μ V for the same day in the absolute values in Fig.S2a. Thus, what it is wrong in the presentation of these data and why are the normalized amplitudes shown as time course in Fig 2B on the left, but the absolute amplitudes used for the statistics displayed in Fig.2 C-E?

Given the similarity of the curves of the absolute VEP amplitudes shown in Fig.S2a and the broad overlap of their confidence band, the statement that visual plasticity is disrupted at this early stage (i.e. 5 months of age) of neurodegeneration should be toned down.

line 134: Again, the normalized values are shown in subfigure B on the left, but the absolute values are used for statistics and shown in C and D. This is not consistent and misleading in the view of the reviewer!

We think there might have been a misunderstanding about what the normalised graphs represent (Fig. 2B, 3B), which is the difference between the VEP amplitude on each day and the VEP amplitude on day 1, calculated for each animal and then averaged within each group. We now make this clear in the figure legend (Fig. 2) and associated text (line 104). Animals have variable day 1 VEP amplitudes, resulting in a large confidence interval of the mean of the absolute VEP values, but show a similar rate of increase over days, therefore reducing the confidence band in the normalised graphs.

Plasticity is assessed as the relative change in the VEP amplitude over days and as we found variable VEP amplitude on day 1 in individual animals, we think that comparing the change in response from day 1 is more meaningful for assessing plasticity between groups. Therefore, we chose to present the change in VEP amplitude from day 1 in main figures (Fig. 2B, 3B). To allow the reader to appreciate the variability between animals, we also included the absolute VEP values in the main (Fig. 2C-E) and supplementary figures (Fig. S5).

To assess significance between the two groups we used repeated measures ANOVA which considers the variability between animals. Specifically, the interaction term (day \times genotype) reveals

whether there is a significant difference in the slope of increase between groups. We then use post hoc tests (Tukey's) to assess whether the VEP amplitude on each day is significantly different to day 1 (Fig. 2C-E) for each group. These tests assess whether there is a significant change from day 1 and therefore no normalisation is needed.

For completeness we now also include posthoc tests that compare the change in the VEP amplitude (normalised values) between Tau+ and Tau- mice on each day.

(R1.8) Figure 4: The scheme in Fig.4A seems to indicate that the individual example for a short-term response of the VEP amplitude from a 5-month-old mouse shown in B below is taken from day 3. If this supposition is not correct, please explain what the dotted lines mean. Furthermore, indicate clearly whether the example for an 8-month-old mouse is also from day 3.

Fig 4B shows the average VEPs across all 5m Tau- and Tau+ animals, for all of days 2-8 (we omitted days 1 and 9, because two orientations were shown in interleaved fashion on these days); Fig 4E shows the equivalent averages for 8m animals. The dotted lines attempt to illustrate that the data in Fig 4C/E are describing within day effects, not across day effects. We have now clarified these points in the figure legend.

(R1.9) line 182: The statistical test used to compare the cumulative histograms of the fitted amplitudes should be given in the Methods and here in the text.

We now note the statistical test (Students t-test) in the legend as well as in the main text.

(R1.10) line 219: You refer here to Fig.4 for within day variations, is this correct?

Correct.

(R1.11) line 230: Based on the data of your elastic net regularization model, you conclude here that behaviour cannot explain the VEP increase across days, supporting the presence of plasticity. For most of the readers this model will be a black box and difficult to assess. Importantly, the analysis showed that the VEP amplitude is dependent on the behavioural state. This applies not only to the basal state but also to plasticity measures (as shown in Fig. S5) because they are usually more sensitive to behavioural modulation than basic electrophysiological readouts. As mentioned already before, I recommend to re-analyse further sets of long-term plasticity data, e.g. the data shown in Figure S5 for differences between Tau- and Tau+ mice.

We have now expanded the text (lines 278-283) and methods section (lines 550-561) to explain the model better.

Lines 550-561: "We fitted an elastic net regularisation model to predict the VEP amplitude (target variable) for each animal using days, block number within each day, movement speed and pupil area as regressors (also known as features or input variables). The model assigns weights to the defined features (regressors) in order to predict the output (VEP amplitude). Elastic net regularisation is preferred over linear regression because it encourages sparse solutions, i.e. solutions where some of the learned variable weight coefficients are equal to 0, improving model generalisation, and hence prediction accuracy. Model sparsity is determined by elastic net's two regularisation hyperparameters. Because we don't know the optimal values of these hyperparameters, we explore the entire regularisation path. Here we used $L \leq 10,000$ values of the regularisation hyperparameter, fixing the ratio of the L1 over L2-norm regularisers to 2. We then estimated the impact of each regressor, based on the derived regression weights, as the percentage of contribution to the estimates of the VEP amplitude across the entire regularization path."

We have also reanalysed the data as the reviewer suggested (see response to R1.1).

Discussion

(R1.12) line 303: “There is some evidence...” Please explain here in more detail how the impairment of inhibitory function early in neurodegeneration may result in reduced SRP in Tau+ animals.

SRP has been shown to depend on the activity of PV+ inhibitory neurons and one hypothesis is that SRP results from lasting homosynaptic long-term depression of glutamatergic input to those neurons. However, as the involvement of inhibition on SRP or tauopathies is not yet clear we have de-emphasised inhibition in our statement (line 359-364).

(R1.13) line 327: As mentioned above, the proven direct effects of Tau pathology on long-term plasticity are rather mild in the current version of the ms. Therefore, phrases like “allows easy characterization of the impact of neurodegeneration on intrinsic plasticity” should be toned down, as long no additional evidence and facts are provided.

Done (line 385).

(R1.14) line 333: In this context, published data about NMDAR function in rTg4510 mice should be mentioned and discussed [e.g. Ma et al. (2020), Curr Med Sci 40:1031-1039].

We now discuss the involvement of NMDA and AMPA receptors in the expression of SRP and their loss as a possible consequence of the observed reduced plasticity in Tau+ mice (lines 420-425).

(R1.15) line 363ff: It is not clear to the reviewer, why increased synaptic instability is chosen here as the prime mechanism to explain the functional deficits observed by the authors in rTg4510 mice at early stages of neurodegeneration.

We toned down our statement on synaptic instability and also discuss the loss of NMDAR and AMPAR as a possible explanation of the functional deficits observed (lines 420-425).

Methods

(R.16) line 422: From the present description of the experiments it is difficult to deduce in detail when which experiments were carried out with which animals and whether certain topics were investigated with sub-data sets from the same experiments. Therefore, please provide a clear diagram of the timeline of the individual experiments.

We changed the legend and panel A in Fig. 1 to clearly indicate that different cohorts of animals were tested at 5 and 8 months.

Reviewer #2:

In this manuscript, Papanikolaou et al. examine the impact of Tauopathy on visual cortical function and plasticity, using the TG4510 mouse model. In freely behaving mice, they assess visual cortical response to a simple, grating stimulus and measure visual cortical plasticity across different timescales as mice become familiar with a specific orientation of stimulus. They measure visual cortical function and plasticity at two different ages, comparing littermate controls with mutant Tau over-expressors at 5 months of age and 8 months of age. They demonstrate that, while basal visual cortical responses seem largely unaffected when measured with electrophysiology, there is nevertheless a compromised behavioural response to a visual stimulus, even in young Tau+ mice. They then reveal that both short-term visual cortical plasticity and long-term plasticity over days is compromised by mutant Tau+ over-expression. Interestingly, this plasticity is generally age-dependent, as it is drastically reduced by 8 months in both Tau- and Tau+ mice. Nevertheless, residual plasticity is also reduced in the Tau+ mice. In general, I think these findings are important

because they demonstrate a means to measure plasticity as an animal learns in disease models. This represents substantial progress over the more standard approach of experimental tests of synaptic plasticity such as LTP and LTD, which may be conducted in vivo or ex vivo but, in both cases, are induced by extremely artificial means. An additional future benefit of this work may be that some similar methods could be applied in human subjects.

We thank the reviewer for pointing out the contribution and importance of our manuscript.

I do have some concerns about the interpretation of the findings in this manuscript that I will now discuss:

(R2.1) The study is designed to assess the impact of neurodegeneration on plasticity, and this is the way that both the abstract and introduction are pitched. However, the ages of the mice that have been selected here are both prior to the documented neurodegeneration threshold of 8.5 months, as stated by the authors in their own introduction. Why not, therefore, include a group that is considerably past this point and with clear neurodegeneration.

The main target of our study was long term plasticity, and our results show that long term plasticity is significantly reduced in older WT and Tau- animals, as well as Tau+ animals. It is likely that long-term plasticity in all animals would be further reduced at ages above 8 months, and as a result differences in between Tau- and Tau+ animals would be too small to test at later ages. We note that previous work shows a significant reduction in the size of cortical areas in 8 month old rTg4510 mice (Fig. 2g in Blackmore et al., 2017) and cortical neurodegeneration by 8.5 months (cortical neuronal loss reaches ~52% by 8.5 months according to Spires et al., 2006). We also see reduction in brain weight in our 8 month cohort of Tau+ animals. We think that our previous wording in our introduction may have implied that there was no degeneration in 8m rTg4510 animals, and we have now reworked to make such a misunderstanding (lines 24-25).

(R2.2) It would be helpful to have some indication of the degree of neurodegeneration/tauopathy in visual cortex of the mice used here. If the authors still have the brains of the mice used, then that could be evaluated. There might be an opportunity to compare the degree of neurodegeneration or tauopathy with the impact on cortical plasticity on a mouse-by-mouse basis. Alternatively, the emphasis of the manuscript could be switched to tauopathy rather than neurodegeneration.

We agree that the major findings in this paper are better framed as impact of tauopathy, rather than neurodegenerations. Therefore, we have rephrased the manuscript to be explicit about this.

Our histological preparations weren't designed to allow us to robustly count neurons. However, Spires et al. (2006) show reduction in cell numbers at 8m but not 5m.

(R2.3) The mismatch between the intact visual cortical response and the lost visual-induced behavioural response may indicate that the behaviors tested here require normal function in other visual pathways/regions, perhaps including the superior colliculus. I suggest it would be worth discussing this possibility or providing a strong justification for the behaviors that have been used as assays of visual cortical function.

We have included a discussion point about the Superior Colliculus being an area potentially affected in this animal model (lines 454-457). However, histological analyses show little accumulation of aberrantly phosphorylated tau (PG5 staining) in Superior Colliculus in these animals (Fig. 5 in Harrison, I. F. et al. *Optic nerve thinning and neurosensory retinal degeneration in the rTg4510 mouse model of frontotemporal dementia. Acta Neuropathol Commun* **7**, 4, doi:10.1186/s40478-018-0654-6 (2019)).

(R2.4) A growing view among those that work on dementia is that synaptic dysfunction precedes neurodegeneration by quite some time and that it may serve as the origin for mild cognitive impairments or even more prodromal symptoms of dementia. Therefore, to me, the most exciting

aspect of the findings in this manuscript is that cortical plasticity deficits manifest in tauopathy could be measured in the living animal (and perhaps eventually human) at a very early age. That may present a biomarker with predictive value for Dementia. This concept could be given some emphasis in the manuscript.

We thank the reviewer for pointing this out. We agree and we have emphasized this contribution more in the manuscript in the introduction (lines 40-41) and our discussion (lines 465-466).

(R2.5) It seems likely that minimal neurodegeneration has occurred by the time these mice are showing this pronounced deficit in plasticity at 5 months. It would be very interesting to see if the deficit is present at a much earlier age of, say, 1 or 2 months. Perhaps the overexpression of mutant tau is causing the synaptic plasticity deficit rather than any degenerative process.

rTg4510 mice accumulate an early tau burden in the cortex by 4 months of age (Ramsden et al., 2005; SantaCruz et al., 2005). We decided to study 5m old animals as at that age there is significant accumulation of tau but no cortical cell loss. We now clarify that better in the manuscript (lines 25-27). In our study, Tau- animals were fed with a doxycycline diet from the age of 2 months. Given that Tau- mice show similar SRP to WT animals we think that tau expression until the age of 2 months does not significantly affect these plasticity measures.

(R2.6) The authors raise the issue of inhibition as a key contributor to stimulus-selective response plasticity and that this may be aberrant in the Tauopathy model. However, as the authors themselves point out, the mutant Tau should be over-expressed under the CaMKII promoter, which should not produce expression in cortical inhibitory neurons. Moreover, a loss of cortical inhibition would surely affect visual cortical response magnitude. In fact, I would expect to see greatly increased VEPs. This issue could do with consideration in the discussion section.

Tau accumulation has been shown to take place predominantly in excitatory neurons compared to inhibitory neurons (Fu et al., 2018) but there is evidence that inhibitory synapses are affected early in tauopathy and before excitatory synapses (Shimojo et al., 2020). Tau+ mice in our study show slightly increased VEPs compared to Tau- mice, that may be an effect of reduced inhibition. However, since this effect was not statistically significant and the involvement of inhibition both in SRP and in tauopathy is not yet clear, we have toned down our statement (lines 359-364).

REVIEWERS' COMMENTS:

Reviewer #1 (Remarks to the Author):

All of my concerns have been thoroughly addressed by the authors, and I have no further comments or questions.

Reviewer #2 (Remarks to the Author):

I thank the authors for responding to all of the comments of the reviewers and modifying the manuscript accordingly. I have no further requests or comments and believe that this manuscript now provides a substantial and important contribution to the field.